# Leveraging Rotation Symmetry for Efficient LoRA Merging in Large Language Models

## Abstract

Merging a large number of low-rank adaptations (LoRAs) is a key technology for enhancing the integration and deployment efficiency of large language models (LLMs). However, this process has long been hindered by the catastrophic "parameter interference" problem, which often leads to a sharp decline in model performance after merging. Existing merging methods are vulnerable when dealing with complex conflicts, such as high-rank LoRAs. While the classical rotation alignment approach can enhance robustness, it is difficult to apply due to incompatibility with the LoRA structure and its high computational complexity. To address these challenges, we propose a novel two-stage parameter alignment (TSPA) framework. TSPA fundamentally overcomes the limitations of existing methods through two core strategies: (1) we innovatively design an alignment mechanism within the LoRA low-rank space, which effectively resolves the structural compatibility issue while maintaining functional equivalence; (2) we introduce an alignment paradigm of "comparison with an average model," which reduces computational complexity from quadratic to linear, ensuring the scalability of the method. To guide the alignment process, TSPA further designs two complementary optimization objectives: macro-functional alignment and micro-parameter alignment, and uses Stiefel manifold optimization to solve the problem, steadily maintaining the orthogonality of the rotation matrices during iterations. We conduct experiments on Natural Language Processing (NLP) tasks using models such as Llama-3-8B. The results show that TSPA not only outperforms state-of-the-art (SOTA) baseline methods, including DARE, in terms of average performance across tasks but also demonstrates unique comprehensive advantages: its two-stage design achieves the optimal balance between task capabilities and general knowledge; it exhibits greater robustness than SOTA methods in high-rank and high-interference scenarios; and it shows significant effectiveness in retaining fine-grained functions, such as "safety capability." This work presents a novel and practical framework for efficient, powerful, and stable multi-task model merging.

## 1 Introduction

The growing demand for specialization in LLMs has led to the emergence of numerous LoRA adapters (Taori et al., 2023; Cui et al., 2023). Merging these LoRA adapters is a crucial step in integrating model capabilities and enhancing deployment efficiency. However, direct merging can result in catastrophic "parameter interference," causing a sharp decline in the performance of the merged model, or even rendering it inoperable. Therefore, effectively mitigating parameter interference is the most critical and urgent challenge in the field of multi-LoRA merging.

To address the issue of parameter interference, a series of attempts have been made in the academic community (Matena & Raffel, 2022; Jin et al., 2022; Ilharco et al., 2022; Yadav et al., 2024; Yu et al., 2024). However, these methods each have notable limitations: (1) Simple linear merging methods, such as Task Arithmetic (Ilharco et al., 2022), often perform poorly, with the merged model's performance sometimes dropping dramatically, rendering it inoperable. (2) More refined pruning or masking methods, such as TIES-Merging (Yadav et al., 2024) and DARE (Yu et al., 2024), show some improvements but expose their vulnerabilities when faced with more complex parameter conflicts (e.g., high-rank LoRA), significantly reducing their effectiveness. On the other hand, the "rotation alignment" approach offers an elegant solution from a geometric perspective

(Zhang et al., 2025). However, when applied to real-world scenarios involving large-scale multi-LoRA merging, it faces two nearly insurmountable obstacles: (1) Structural incompatibility: The target of the rotation operation is the complete weight matrix, which disrupts the low-rank structure that LoRA relies on for success, rendering it inapplicable. (2) Excessive computational complexity: This method requires pairwise comparisons of models, causing the computational cost to grow quadratically ($O(n^2)$) with the number of models $n$, making it infeasible in scenarios where dozens of LoRAs need to be merged. These limitations of existing methods lead us to reflect and propose the core research question of this paper: How can we design a multi-LoRA rotation alignment method that is both computationally efficient (linear complexity) and structurally compatible (preserving the LoRA characteristics)?

To address the challenges outlined above, we propose a novel **T**wo-**S**tage **P**arameter **A**lignment (TSPA) framework designed to achieve efficient and robust multi-LoRA merging. Specifically, TSPA is built upon two core strategies, each addressing the challenges of structure and efficiency: (1) To resolve structural compatibility issues, we innovatively design an alignment mechanism within a low-rank space. This strategy performs geometric rotations of the LoRA's A and B matrices in pairs, ensuring full structural compatibility while preserving their original functional equivalence. (2) Building on this mechanism, we further propose an alignment paradigm that leverages comparison with the average model to enhance computational efficiency. This strategy discards the inefficient "pairwise comparison" approach of traditional methods, successfully reducing the computational complexity from quadratic ($O(n^2)$) to linear ($O(n)$). These strategies provide the technical and computational feasibility for alignment, but a more crucial question remains: in what direction should we align? To answer this, we must define a final objective to measure the merging outcome. Therefore, to guide the alignment process toward capability fusion rather than conflict, we further design two complementary optimization objectives. The first is macroscopic functional alignment, which involves aligning the attention score matrices of each model. The second is microscopic parameter alignment, focusing on aligning the LoRA update matrices of each model. To solve this complex, constrained optimization problem, we model it as an optimization task on the Stiefel manifold, based on the fundamental requirements of optimization theory. This ensures that we can efficiently solve it using modern gradient-based optimizers (such as AMSGrad (Reddi et al., 2019)), while strictly maintaining the orthogonality of the rotation matrices at each iteration (Bonnabel, 2013; Bécigneul & Ganea, 2018). Consequently, we present the complete TSPA framework: it addresses both the technical and efficiency bottlenecks with two core strategies, directs the merging process with two optimization objectives, and ensures the correctness of the solution through manifold optimization, ultimately achieving powerful and stable multi-LoRA model fusion.

To systematically evaluate the effectiveness of the TSPA framework, we conduct comprehensive experiments comparing our method with several SOTA baseline methods, including TIES-Merging (Yadav et al., 2024) and DARE (Yu et al., 2024), on models such as Llama-3-8B (AI@Meta, 2024). The experimental results demonstrate that TSPA effectively alleviates the severe parameter interference caused by multi-LoRA merging. Through an in-depth analysis of merging scenarios with different numbers of models and various LoRA configurations, we derive the following key findings:

1. TSPA addresses two major bottlenecks of existing rotation alignment methods: structural compatibility and computational efficiency. It achieves favorable compatibility with the LoRA structure through an innovative low-rank space alignment mechanism. Additionally, its design with linear complexity ensures efficiency even in large-scale merging scenarios.

2. TSPA's two-stage design achieves a balance between functional and parameter alignment. After resolving the fundamental bottlenecks, our ablation experiments further confirm that while pure functional alignment excels in specific tasks, it sacrifices general capabilities. Conversely, pure parameter alignment achieves a balanced performance but lacks top-tier effectiveness. TSPA integrates both, resulting in a more comprehensive and powerful overall performance.

3. TSPA demonstrates stronger robustness than existing SOTA methods under high-intensity interference scenarios. Notably, in the high-rank (rank=32) LoRA merging experiment, methods like TIES-Merging experience performance collapse, while TSPA maintains robust and excellent results, proving its superiority in handling complex parameter interference.

4. TSPA excels in preserving delicate and fragile model functionalities, such as "safety capabilities." In all experiments, TSPA consistently outperforms other methods in safety evaluations, indicating

that our alignment strategy is better at protecting fine-grained parameter structures that are most vulnerable during merging.

These findings collectively demonstrate that TSPA is not only an effective LoRA merging solution but also an advanced framework with significant advantages in robustness, balance, and the protection of delicate capabilities. It offers a new approach for achieving powerful multi-task AI models.

## 2 RELATED WORK

### 2.1 MODEL MERGING

Model merging focuses on combining the weights of models fine-tuned for various downstream tasks, resulting in a more capable model while reducing additional storage overhead. The simple averaging method (Wortsman et al., 2022) directly merges the model parameters but often leads to a significant performance degradation. In contrast, the Fisher-Merging (Matena & Raffel, 2022) and RegMean (Jin et al., 2022) methods leverage the Fisher information matrix and the inner product matrix of the training data, respectively, to compute more effective linear weighting coefficients. The mainstream approaches now rely on a paradigm introduced in Task Arithmetic (Ilharco et al., 2022) to facilitate both task forgetting and learning. The core idea behind this approach is to merge the task vectors (i.e., the parameter differences between fine-tuned and pre-trained models), rather than the entire weight matrices, enabling more flexible adjustments to the model weights. TIES-Merging (Yadav et al., 2024) addresses the key issue of performance degradation following linear weighting by highlighting interference between the parameters of different models. It mitigates this issue by trimming redundant parameters and selecting the appropriate sign for each parameter, reducing the impact of interference. The DARE method (Yu et al., 2024), on the other hand, offers a parameter sparsification preprocessing technique that alleviates parameter interference through random dropping and rescaling operations. In addition to full fine-tuning, an increasing number of researchers are utilizing LoRA for fine-tuning downstream tasks, achieving impressive results by adjusting only a small number of parameters. EMR-Merging (Huang et al., 2024) aims to elect a unified model and align the direction and magnitude of the unified model with the original task-specific models. PCB-Merging (Du et al., 2024) measures parameter significance within individual tasks using intra-balancing and evaluates parameter similarities across different tasks using inter-balancing. The concept of LoRA merging has emerged as a result, with existing studies demonstrating that combining LoRA adapters through arithmetic operations leads to a more powerful model (Zhang et al., 2023). KnOTS (Stoica et al., 2024) further introduces singular value decomposition (SVD) to transform different LoRA models into an aligned space, while remaining compatible with existing approaches. RobustMerge (Zeng et al., 2025) focuses primarily on LoRA merging for multimodal large language models and proposes a method for adaptively scaling tail singular values.

### 2.2 PARAMETER SPACE SYMMETRY

Parameter space symmetry refers to a property of a set of models that, despite having different parameters, maintain functional equivalence. One significant application of this concept is linear mode connectivity (LMC), which demonstrates that local optima in the loss landscape of deep neural networks can be connected by simple curves (Garipov et al., 2018; Draxler et al., 2018). A prior work explores the relationship between linear mode connectivity and the lottery ticket hypothesis from the perspective of model training (Frankle et al., 2020). For model merging, finding symmetric models that possess the properties of LMC offers a solution to mitigating parameter interference between models. Permutation symmetry aligns the models of two MLP layers using permutation matrices, ensuring that the models after permutation satisfies LMC, so that the performance of the model after linear combination does not significantly degrade compared to the original models (Ainsworth et al., 2022). However, this alignment strategy is no longer applicable to models based on the Transformer architecture (Vaswani et al., 2017). Therefore, rotation symmetry is proposed as a method to align the two attention layers in continuous space using rotation matrices (Zhang et al., 2025). The determination of the appropriate rotation matrices is equivalent to the Orthogonal Procrustes problem (Schönemann, 1966; Gower & Dijksterhuis, 2004), for which a closed-form solution exists. Nonetheless, this approach cannot be directly applied to LoRA merging. To address this, we extend rotation symmetry to align the attention layers of multiple models, while also introducing align-

ment operations within the internal matrices of the LoRA adapters to further alleviate parameter interference.

## 3 METHOD

### 3.1 INTUITION

Model merging integrates multiple models into a single, more powerful model. However, due to parameter interference between different models, the performance of the merged model is often inferior to that of the original fine-tuned models (Yadav et al., 2024). As illustrated in Figure 1, models A, B, and C are situated near distinct local optima. Without alignment, the parameters of the merged model may fall into a non-local optimum, resulting in a substantial performance decline. However, after parameter alignment, models A and B are mapped to A' and B', which can be positioned near the same local optimum as C, thereby enabling superior performance when merged. Parameter alignment geometrically projects the models onto the vicinity of the same local optimum while ensuring functional equivalence through the symmetry properties of the models. Consequently, parameter alignment ensures that the aligned models possess LMC characteristics, enabling the merged models to maintain outstanding performance.

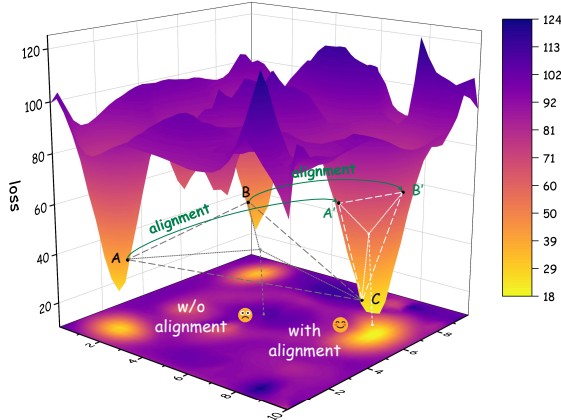

Figure 1: An intuitive example of a 3D surface plot and the corresponding contour map of the loss landscape in the parameter space. Colors closer to light yellow indicate lower model loss, while colors closer to dark purple correspond to higher model loss. A, B, and C represent three original fine-tuned models, each located near different local optima within the loss landscape. A' and B' are the equivalents of A and B after parameter alignment. (Figure placement revised.)

### 3.2 ROTATION SYMMETRY

Parameter space symmetry refers to the property of a set of models with identical functionality but different parameters, where the parameters can be transformed into each other through a certain projection operation. One such type is permutation symmetry, which aligns the parameters of MLPs using permutation matrices (Ainsworth et al., 2022). These permutation matrices can be propagated layer by layer, so that the problem essentially a combination of multiple bilinear assignment problems. Rotation symmetry, on the other hand, brings the characteristics of permutation symmetry into continuous space by using rotation matrices to align the attention layers (Zhang et al., 2025). In the following, we introduce the rotation matrices in the context of the query and key matrices at a specific layer of the model, with similar conclusions for the value and output matrices. Let these matrices be denoted as $\mathbf{W}_Q$ and $\mathbf{W}_K$, respectively, and let the rotation matrix be denoted by $\mathbf{R}$, which satisfies $\mathbf{R} \cdot \mathbf{R}^T = \mathbf{I}$. As a result, the following relation holds:

$$(\mathbf{W}_Q \mathbf{R})(\mathbf{W}_K \mathbf{R})^T = \mathbf{W}_Q \mathbf{W}_K^T \tag{1}$$

Therefore, the orthogonality of the matrices ensures the invariance of the model's functionality after the transformation. The specific solution for the two models can be obtained by optimizing the objective function given in:

$$\min_{\mathbf{R}_1, \mathbf{R}_2 \in \mathcal{R}} \left\| [\mathbf{W}_{Q_1} \quad \mathbf{W}_{Q_2}] \begin{bmatrix} \mathbf{R}_1 \\ -\mathbf{R}_2 \end{bmatrix} \right\|_F^2 + \left\| [\mathbf{W}_{K_1} \quad \mathbf{W}_{K_2}] \begin{bmatrix} \mathbf{R}_1 \\ -\mathbf{R}_2 \end{bmatrix} \right\|_F^2 \tag{2}$$

where $\mathbf{W}_{Q_1}$ and $\mathbf{W}_{Q_2}$ represent the query matrices for the two models, while $\mathbf{W}_{K_1}$ and $\mathbf{W}_{K_2}$ represent the key matrices for the two models. $\mathbf{R}_1$ and $\mathbf{R}_2$ are the corresponding rotation matrices. This problem is essentially an Orthogonal Procrustes problem (Schönemann, 1966; Gower &

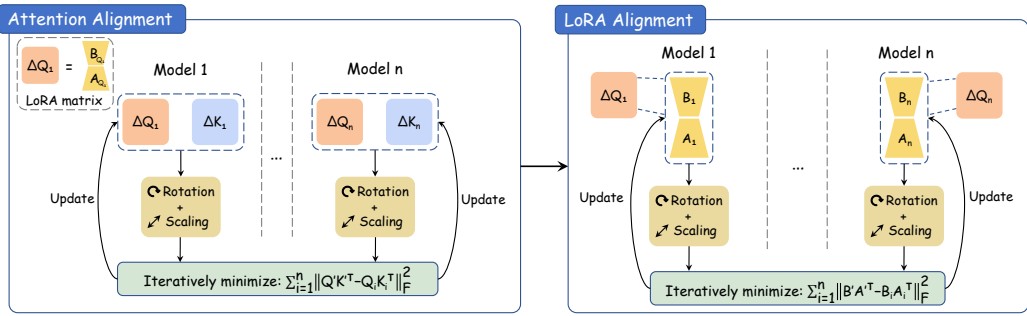

Figure 2: Framework of the two-stage parameter alignment process. For simplicity, we denote the query and key matrices as $\mathbf{Q}'$ and $\mathbf{K}'$, rather than $\mathbf{W}'_Q$ and $\mathbf{W}'_K$. Attention alignment focuses on comparing the changes in the query-key matrix product, while LoRA alignment examines the variations in the attention matrix, specifically the product of the LoRA matrices B and A. The main idea is to iteratively solve for the rotation matrices and scaling factors, ensuring that the model retains equivalent functionality after alignment, while minimizing interference caused by linear weighted merging operations. (Figure placement revised.)

Dijksterhuis, 2004), and its closed-form solution is given by:

$$\mathbf{R}_1 = \mathbf{U}\mathbf{V}^T, \mathbf{R}_2 = \mathbf{I} \tag{3}$$

where $\mathbf{U}\mathbf{\Sigma}\mathbf{V}^T = \mathbf{W}_{Q_2}^T \mathbf{W}_{Q_1} + \mathbf{W}_{K_2}^T \mathbf{W}_{K_1}$ represents the singular value decomposition, and $I$ is the identity matrix.

The rotation alignment between two models can be easily computed using a closed-form solution, and by extend Equation (2) to multi-model merging, we obtain the following objective function:

$$\min_{\mathbf{R}_i, \mathbf{R}_j \in \mathcal{R}} \sum_{i=1}^{n} \sum_{j=i+1}^{n} \left\| \begin{bmatrix} \mathbf{W}_{Q_i} & \mathbf{W}_{Q_j} \end{bmatrix} \begin{bmatrix} \mathbf{R}_i \\ -\mathbf{R}_j \end{bmatrix} \right\|_F^2 + \left\| \begin{bmatrix} \mathbf{W}_{K_i} & \mathbf{W}_{K_j} \end{bmatrix} \begin{bmatrix} \mathbf{R}_i \\ -\mathbf{R}_j \end{bmatrix} \right\|_F^2 \tag{4}$$

However, applying this form to the practical scenario of large-scale multi-LoRA merging presents two nearly insurmountable obstacles: (1) Structural incompatibility: The merged incremental matrices obtained from Equation (4) are full-weight matrices, which completely disrupt the low-rank structure that LoRA methods rely on for success, making direct application unfeasible. (2) Excessive computational complexity: As the number of models increases to $n$ ($n > 2$), the result no longer has a closed-form solution, and the computational cost grows quadratically ($O(n^2)$). This becomes entirely infeasible in scenarios that require merging dozens of LoRAs. Due to these two significant challenges, the application of rotation symmetry in multi-LoRA merging is constrained, prompting us to conceive and design a computationally efficient (linear complexity) and structurally compatible (preserving LoRA characteristics) multi-LoRA rotation alignment method.

### 3.3 TWO-STAGE PARAMETER ALIGNMENT

We propose a novel two-stage parameter alignment framework (TSPA) designed to enable efficient and effective multi-LoRA merging, as illustrated in Figure 2. Specifically, TSPA relies on two core strategies, each designed to address the challenges of structural compatibility and efficiency. To tackle structural incompatibility, we introduce an innovative alignment mechanism within a low-rank space. This strategy transforms the LoRA A and B matrices in pairs, performing geometric rotations and scaling operations that are fully compatible with the LoRA structure, while maintaining functional equivalence of the model. To address the issue of computational efficiency, we propose an alignment paradigm based on comparison with the average model. This strategy discards the inefficient "pairwise comparison" approach used in traditional methods, successfully reducing the computational complexity from quadratic ($O(n^2)$) to linear ($O(n)$). These strategies ensure the feasibility of TSPA both technically and computationally. However, more crucially, we require a objective function similar to Equation (4) to guide the alignment direction, steering the alignment process towards capability fusion rather than conflict. We further design two complementary optimization steps: (1) Attention Alignment: This step involves macro-level functional alignment,

aligning the attention score matrices across models. (2) LoRA Alignment: This step focuses on micro-level parameter alignment, aligning the LoRA update matrices across models. To solve this complex, constrained optimization problem, we base our approach on the fundamental requirements of optimization theory and model it as an optimization task on the Stiefel manifold (a type of Riemannian manifold). This ensures that we can efficiently solve the task using gradient-based modern optimizers (e.g., Adam (Kingma & Ba, 2014) and AMSGrad (Reddi et al., 2019)), while rigorously preserving the orthogonality of the rotation matrices at each iteration (Bonnabel, 2013; Bécigneul & Ganea, 2018). The related code implementation and detailed comments can be found in the supplementary material. A detailed explanation of the two stages of parameter alignment in TSPA is provided below.

**Attention alignment.** We treat the linearly merged model as the average model and calculate the parameter distance between it and the original models. Since LoRA merging requires applying linear combination to the LoRA matrices A and B separately, the merged form, taking the query and key matrices as an example, is expressed as:

$$\mathbf{W}_Q^{'} = \mathbf{W}_Q + \left( \sum_{i=1}^{n} \alpha_i \mathbf{B}_{Q_i} \right) \left( \sum_{i=1}^{n} \alpha_i s_i \mathbf{A}_{Q_i} \mathbf{R}_i \right) \tag{5}$$

$$\mathbf{W}_K^{'} = \mathbf{W}_K + \left( \sum_{i=1}^{n} \alpha_i \mathbf{B}_{K_i} \right) \left( \sum_{i=1}^{n} \frac{\alpha_i}{s_i} \mathbf{A}_{K_i} \mathbf{R}_i \right) \tag{6}$$

where $\mathbf{B}_{Q_i}$ and $\mathbf{A}_{Q_i}$ are the LoRA matrices for the query matrix of the $i$-th model, $\mathbf{B}_{K_i}$ and $\mathbf{A}_{K_i}$ are the LoRA matrices for the key matrix of the $i$-th model, and $\alpha_i$ is the linear weighting coefficient for the $i$-th model during LoRA merging. $s_i$ denotes the scaling factor of the matrix for the $i$-th model, such that the attention score matrices for the $i$-th model remains unchanged (Neyshabur et al., 2015; Meng et al., 2018). $\mathbf{R}_i$ represents the rotation matrix of the $i$-th model, while during LoRA merging, the rotation matrix is applied to the LoRA A matrix. The objective function in Equation (4) heuristically compares the distances within the query matrices and within the key matrices separately. However, this approach overlooks the changes in the merged model's attention score matrices, specifically the product of the query and key matrices. Therefore, in pursuit of macroscopic functional alignment, we design the following objective function based on the attention score matrices:

$$\min_{\mathbf{R}_i, \mathbf{R}_j \in \mathcal{R}} \sum_{i=1}^{n} \left\| \mathbf{W}_Q^{'} \mathbf{W}_K^{'T} - \mathbf{W}_{Q_i} \mathbf{W}_{K_i}^{T} \right\|_F^2 \tag{7}$$

By aligning with the "average model," we successfully reduce the original quadratic complexity ($O(n^2)$) to linear complexity ($O(n)$).

**LoRA alignment.** The goal of LoRA alignment is to perform parameter alignment at the micro level, specifically aligning the LoRA update matrices within the model's attention matrices. Taking the query matrix as an example, the LoRA matrices of the merged model are given by:

$$\mathbf{B}_{Q_i}^{'} = \sum_{i=1}^{n} \alpha_i s_{Q_i} \mathbf{B}_{Q_i} \mathbf{R}_{Q_i}, \quad \mathbf{A}_{Q_i}^{'} = \sum_{i=1}^{n} \frac{\alpha_i}{s_{Q_i}} \mathbf{R}_{Q_i}^{T} \mathbf{A}_{Q_i} \tag{8}$$

where $s_{Q_i}$ represents the scaling factor of the query matrix in the $i$-th model. Therefore, the objective function is:

$$\min_{\mathbf{R}_{Q_i}, \mathbf{R}_{Q_j} \in \mathcal{R}} \sum_{i=1}^{n} \left\| \mathbf{B}_Q^{'} \mathbf{A}_Q^{'} - \mathbf{B}_{Q_i} \mathbf{A}_{Q_i} \right\|_F^2 \tag{9}$$

Similar to the iterative solution of Equation (4) in attention alignment, we also need to iteratively solve for the rotation matrices and scaling factors to ensure the alignment of parameters at a microscopic level.

## 4 EXPERIMENTS

### 4.1 EXPERIMENTAL SETTINGS

**Fine-tuning configuration.** To evaluate the effectiveness of multi-LoRA merging, we fine-tune the base model Llama-3-8B (AI@Meta, 2024) on four widely used generative tasks in the NLP

| Merging Methods | General Capabilities | | Instruction-Following | Math | Code | Safety | Avg. |
|---|---|---|---|---|---|---|---|
| | MMLU | TriviaQA | IFEval | GSM8K | HumanEval | DirectHarm4 | |
| Base | 62.19 | 63.34 | 21.34 | 38.36 | 35.37 | 29.25 | 41.64 |
| Fine-tuned | / | / | 28.42 | 43.82 | 41.46 | 89.75 | / |
| Task Arithmetic | 22.95 | 0.00 | 25.06 | 0.00 | 0.00 | 0.00 | 8.00 |
| TIES-Merging | 62.21 | 62.67 | 26.02 | 48.37 | 38.41 | 54.25 | 48.66 |
| DARE+TIES-Merging | _62.31_ | **63.16** | 25.78 | 48.82 | 36.59 | 57.00 | 48.94 |
| EMR-Merging | 22.95 | 0.00 | 0.00 | 38.44 | 35.37 | 28.75 | 20.92 |
| PCB-Merging | **62.37** | 62.68 | 27.82 | 46.40 | 36.59 | 59.75 | 49.27 |
| KnOTS+TIES | 60.50 | 54.33 | **41.01** | **49.73** | **42.68** | 58.00 | 51.04 |
| **TSPA(Attention)** | 61.99 | 57.82 | _29.62_ | 48.75 | 39.63 | **80.25** | **53.01** |
| **TSPA(LoRA)** | 61.29 | _62.86_ | 28.78 | 45.34 | 37.20 | 64.25 | 49.95 |
| **TSPA** | 62.16 | 62.78 | 26.74 | _49.43_ | _40.85_ | _72.50_ | _52.41_ |

Table 1: Test results of the four models, after being merged, each fine-tuned on Instruction-Following, Math, Code, and Safety tasks. Bold and underlined text represent the optimal and sub-optimal results, respectively.

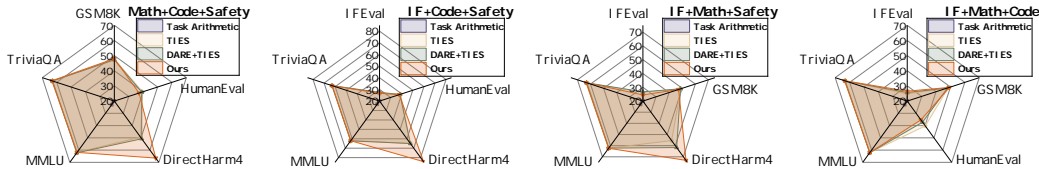

Figure 3: Radar chart of performance for merging three models. The models are selected from the Instruction-Following, Math, Code, and Safety tasks for combination testing, resulting in four possible combinations. (Figure revised.)

domain: Instruction-Following, Math, Code, and Safety. The fine-tuning process utilizes the following datasets: Alpaca-cleaned (Taori et al., 2023), NuminaMath-CoT (LI et al., 2024), Code Alpaca (Chaudhary, 2023), and Saferpaca (Bianchi et al., 2023), all of which are fine-tuned using the LoRA method (Hu et al., 2022) within the LLaMA-Factory framework (Zheng et al., 2024). We fine-tune the model using a rank of 8 in the LoRA configuration. Additionally, we conduct supplementary experiments with ranks of 16 and 32. We follow the original LoRA setup and fine-tune the query and value matrices (Hu et al., 2022). Similarly, we fine-tune all the query, key, value, and output matrices as a supplementary experiment. The details of the fine-tuning parameters and hardware and software configurations can be found in Appendix A, while the training logs are provided in the supplementary material.

**Testing configuration.** To systematically validate the effectiveness of the TSPA framework, we compare it with several SOTA baseline methods, including Task Arithmetic (Ilharco et al., 2022), TIES-Merging (Yadav et al., 2024), and DARE (Yu et al., 2024) (used with TIES-Merging). We do not include KnOTS in the comparison, as its performance shows no significant variation across different linear weighting coefficients. Further theoretical and empirical discussions can be found in Appendix B. A thorough analysis is conducted across four models, three models, and different LoRA configurations. We evaluate the general capabilities of the merged model using the MMLU (Hendrycks et al., 2021b;a) and TriviaQA (Joshi et al., 2017) benchmarks. For task-specific evaluations, we utilize IFEval (Zhou et al., 2023), GSM8K (Cobbe et al., 2021), and HumanEval (Chen et al., 2021) to assess the model's performance on Instruction-Following, Math, and Code tasks, respectively. These evaluations were carried out using the lm-eval framework (Gao et al., 2024). For safety evaluation, we use the DirectHarm4 dataset (Lyu et al., 2024) to assess the safety of the generated responses, with the evaluation conducted using the Llama-Guard-3-8B model (Llama Team, 2024). We use linear weighting coefficients of 1.0 in the merging method. To ensure fairness in the tests, we conduct additional experiments where the linear weighting coefficient is varied from 0.5 to 1.0 in increments of 0.1. Considering that performing a grid search for coefficients across all models could lead to dimensional explosion, we apply the same coefficient to all models in each

test. The other parameters of the test and the hardware and software configurations are provided in Appendix A.

## 4.2 MAIN RESULTS

**Merging of four models.** The results of merging the four models are presented in Table 1. When we directly apply Task Arithmetic (Ilharco et al., 2022) to linearly combine the weight matrices, the performance significantly degrades due to most responses becoming garbled. This indicates significant parameter interference between the fine-tuned models. However, after aligning the model parameters using our TSPA framework before performing the linear combination, performance improves, retaining capabilities across the four downstream tasks. Additionally, we separately test the cases involving only attention alignment and only LoRA alignment, denoted as TSPA(Attention) and TSPA(LoRA) in Table 1. Only the attention alignment approach achieves exceptional performance across the four downstream tasks, with notable improvements in instruction-following and safety refusal capabilities, far surpassing other methods. However, its general capabilities, particularly in the TriviaQA task, show a significant decline. On the other hand, the model that only adopts LoRA alignment before merging exhibits more consistent performance across various tasks, with its average results surpassing those of the baseline methods, although it is not top-performing. TSPA outperforms both the TIES-Merging approach (Yadav et al., 2024) and the DARE method (Yu et al., 2024) across the four downstream tasks, although it slightly trails in terms of general capabilities. TSPA avoids the significant decline in general capabilities observed with attention alignment only, while outperforming LoRA alignment only in the Math, Code, and Safety tasks. This demonstrates the necessity of the two-stage parameter alignment within TSPA. By integrating both stages, TSPA achieves more comprehensive and enhanced overall performance. In general, TSPA demonstrates superior average performance compared to the baseline, excelling in delicate and vulnerable model functions such as "safety capabilities." In contrast to Task Arithmetic, which directly performs a linear combination on task vectors, leading to significant performance degradation, the linear combination after parameter alignment through TSPA demonstrates outstanding performance. Therefore, TSPA mitigates parameter conflicts effectively.

**Merging of three models.** We select three models from the four LoRA adapters for merging, resulting in four distinct scenarios, which are illustrated in Figure 3 using radar charts. The performance of Task Arithmetic remains poor, with the majority of output results being garbled. In contrast, after applying TSPA, a significant improvement in outcomes is observed across all merging scenarios. Notably, in terms of safety, TSPA consistently outperforms other methods, suggesting that our alignment strategy is more effective in preserving the delicate parameter structures that are most vulnerable during merging. For other tasks, the performance of TSPA is comparable to that of TIES-Merging and DARE, demonstrating the strong applicability and efficiency of TSPA.

## 4.3 SUPPLEMENTARY EXPERIMENTS

To further validate the effectiveness and robustness of the TSPA framework, we adjust several parameters and configurations during the fine-tuning process to evaluate its performance across different scenarios. For clarity in visualization, we present the average performance of the models across six tasks, with the final results shown in Figure 4a. In this context, "Fine-tuned" refers to the average of the highest values across the four fine-tuned models in each dimension, thus representing an upper bound of performance.

**Different LoRA settings.** We examine the rank settings in LoRA fine-tuning, including different ranks, various LoRA matrices, and a different base model. In contrast to the rank of 8 used in Table 1 and Figure 3, we also test scenarios with ranks of 16 and 32. When the rank is set to 16, the performance of TSPA is comparable to that of other baseline methods. However, when facing more complex parameter interference, the performance of the TIES and DARE methods deteriorates as the rank increases to 32, while TSPA continues to maintain robust and outstanding performance, demonstrating its superiority in handling complex parameter interference. Additionally, we test the scenario in which all attention matrices (i.e., query, key, value, and output matrices) are fine-tuned using LoRA, as depicted in the third section of Figure 3. In this case, the average performance of TSPA remains slightly higher than that of the baseline methods. Moreover, we evaluate the effectiveness of our method on the Qwen3-8B model (Team, 2025), using the same fine-tuning

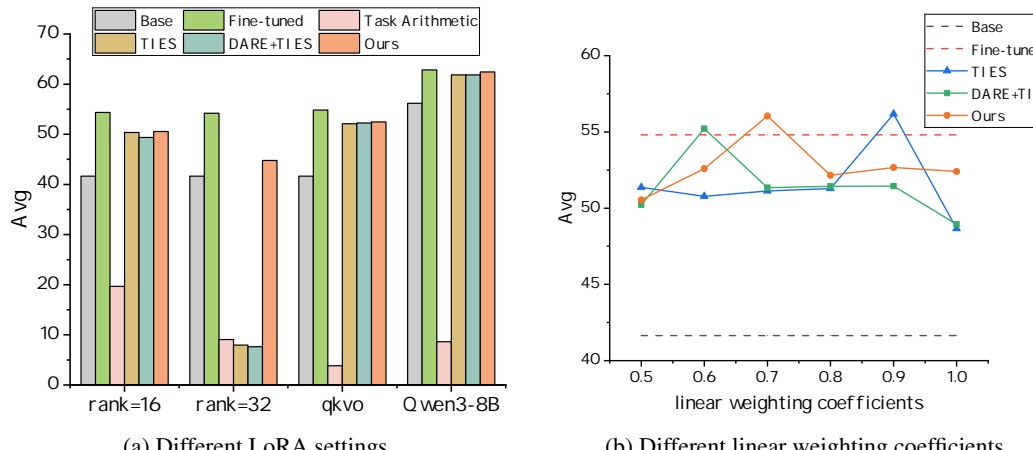

(a) Different LoRA settings.      (b) Different linear weighting coefficients.

Figure 4: (a) Results under different LoRA settings. The horizontal axis represents four different test scenarios: rank=16, rank=32, fine-tuning of all query, key, value, and output matrices, and fine-tuning based on Qwen3-8B (Team, 2025). The vertical axis represents average performance, where the "Fine-tuned" value is the average of the optimal values of the four fine-tuned models across six benchmarks, thus indicating an upper bound. (b) Results under different linear weighting coefficients. We test the average performance of various methods with a step change of 0.1 in the range from 0.5 to 1.0. The results for the base model and the upper bound performance of the fine-tuned models are shown with dashed lines, while the other methods are represented by solid lines.

settings as those in Table 1. As shown in the fourth section of Figure 4a, TSPA similarly mitigates parameter interference and outperforms the baseline methods, even approaching the upper bound performance. This demonstrates the versatility of TSPA across different models.

**Different linear weighting coefficients.** In the previous tests, we use a linear weighting coefficient of 1.0 for all models to avoid the dimensional explosion caused by grid search across the four models. Therefore, we test the scenario under different linear weighting coefficients to evaluate the robustness of TSPA. Due to the poor performance of Task Arithmetic, we omit this method from the figure, and the results are shown in Figure 4b. Interestingly, each method exhibits a scenario with particularly strong performance, highlighting that the optimal coefficient selection varies across methods. Under specific combinations of coefficients, the performance approaches or even exceeds the theoretical upper bound (represented by the dashed line labeled "Fine-tuned" in the figure). In other scenarios, TSPA generally outperforms the baseline methods, demonstrating its robust applicability. As shown in Equation (5), Equation (6), and Equation (8), the design of TSPA incorporates linear weighting coefficients from the outset, enabling it to naturally support different combination coefficients while achieving outstanding performance.

Details on the wall-clock time evaluation, as well as the experiments with different random seeds and the LoRA variant method (DoRA (Liu et al., 2024)), are provided in Appendix C.

## 5 CONCLUSION

In this paper, we propose a novel two-stage parameter alignment framework, TSPA, which extends the application of rotation symmetry in multi-LoRA merging, significantly alleviating the issue of parameter interference. TSPA fundamentally addresses two major bottlenecks of existing methods: incompatibility with the LoRA structure and excessive computational complexity. Specifically, we resolve the structural compatibility issue by designing an alignment mechanism in a low-rank space, which preserves the original functionality equivalently. Moreover, by introducing an alignment paradigm based on "comparison with an average model," we reduce the computational complexity from quadratic to linear, ensuring the scalability of the method. To this end, we design two optimization objectives to achieve macroscopic functional alignment and microscopic parameter alignment. We iteratively optimize these using the Stiefel manifold, with strict preservation of the orthogonality of the rotation matrices throughout the iteration process. Comprehensive experiments are conducted

on Llama-3-8B, evaluating the performance of four-model and three-model merging, as well as testing the robustness of TSPA under different settings. The results show that TSPA outperforms SOTA methods in terms of average performance across various tasks and approaches the upper-bound performance. Additionally, in high-rank and high-interference scenarios, TSPA demonstrates greater robustness than SOTA methods, retaining task capabilities while significantly mitigating the parameter interference issue.

## REPRODUCIBILITY STATEMENT

We are committed to ensuring the reproducibility of our work. The source code and runtime environment configuration are provided in the supplementary material. Our experiments are based on fine-tuned open-source models, with the fine-tuning hyperparameters detailed in Appendix A, and the corresponding training logs are provided in the supplementary material. The datasets and evaluation benchmarks used in our study are all publicly available, as described in Section 4.1.

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

APPENDIX

## A    EXPERIMENTAL SETUP DETAILS

**Hardware and software configuration.** We present the hardware and software configuration details used during fine-tuning and testing in Table 2.

**Hyperparameter Selection.** Regarding the learning rates and hyperparameters utilized in the TSPA framework, we empirically set the learning rates for the attention alignment and LoRA alignment stages to $1e - 1$ and $5e - 2$, respectively, in order to balance training time and performance. The number of training steps is set to 100 and 2000 for the two stages, respectively. The parameter configuration for fine-tuning Llama-3-8B with LoRA on the Alpaca-cleaned, NuminaMath-CoT, Code Alpaca, and Saferpaca datasets is shown in Table 3.

**Random seed configuration.** The random seed used in PyTorch within the TSPA framework is uniformly set to 42. For the fine-tuning process using LLaMA-Factory and the evaluation with the lm-eval framework, we adopt the default random seed settings.

| Device Information | Details |
|---|---|
| Operating System | Ubuntu 22.04.5 LTS |
| CPU Model | Intel Xeon Platinum 8358 @ 2.60GHz (32 cores, 64 threads) |
| CPU Architecture | x86_64 |
| Memory Size | 1.0 TB |
| GPU Model | 8 x NVIDIA A800-SXM4 80GB |
| GPU Memory Size | 640 GB (80 GB per GPU) |

Table 2: Hardware and software configuration information during fine-tuning and testing.

| Dataset | Configuration | Value | Dataset | Configuration | Value |
|---|---|---|---|---|---|
| Alpaca-cleaned | lora_rank | 8 | Code Alpaca | lora_rank | 8 |
| | lora_alpha | 16 | | lora_alpha | 16 |
| | cutoff_len | 2048 | | cutoff_len | 2048 |
| | learning_rate | 2.0e-5 | | learning_rate | 1.0e-4 |
| | num_train_epochs | 3.0 | | num_train_epochs | 3.0 |
| | lr_scheduler_type | cosine | | lr_scheduler_type | cosine |
| | warmup_ratio | 0.05 | | warmup_ratio | 0.1 |
| NuminaMath-CoT | lora_rank | 8 | Saferpaca | lora_rank | 8 |
| | lora_alpha | 16 | | lora_alpha | 16 |
| | cutoff_len | 2048 | | cutoff_len | 2048 |
| | learning_rate | 1.0e-4 | | learning_rate | 1.0e-4 |
| | num_train_epochs | 6.0 | | num_train_epochs | 10.0 |
| | lr_scheduler_type | cosine | | lr_scheduler_type | cosine |
| | warmup_ratio | 0.1 | | warmup_ratio | 0.01 |

Table 3: Main parameters used for fine-tuning with LoRA.

## B    DISCUSSION ON THE KNOTS METHOD

KnOTS (Stoica et al., 2024) introduces a SVD operation in the process of model merging. First, the LoRA delta matrices from a given layer (denoted as $\boldsymbol{\Delta W}_1, \ldots, \boldsymbol{\Delta W}_n$) are concatenated and then decomposed via SVD, yielding:

$$[\boldsymbol{\Delta W}_1 \cdots \boldsymbol{\Delta W}_n] = \mathbf{U}\boldsymbol{\Sigma} \left[ \mathbf{V}_1^T \cdots \mathbf{V}_n^T \right] \tag{10}$$

Subsequently, the SVD components $\mathbf{V}_1^T, \ldots, \mathbf{V}_n^T$ are combined using methods such as Task Arithmetic (Ilharco et al., 2022), TIES-Merging (Yadav et al., 2024), or DARE (Yu et al., 2024) to obtain $\mathbf{V}^{'T}$. The merged matrix is then reconstructed as $\boldsymbol{\Delta W}^{'} = \mathbf{U}\boldsymbol{\Sigma}\mathbf{V}^{'T}$. This computation involves a

| $\alpha$ | General Capabilities | | Instruction-Following | Math | Code | Safety | Avg. |
|---|---|---|---|---|---|---|---|
| | MMLU | TriviaQA | IFEval | GSM8K | HumanEval | DirectHarm4 | |
| *Base* | | | | | | | |
| / | 62.19 | 63.34 | 21.34 | 38.36 | 35.37 | 29.25 | 41.64 |
| *Fine-tuned* | | | | | | | |
| / | / | / | 28.42 | 43.82 | 41.46 | 89.75 | / |
| *KnOTS+TIES-Merging* | | | | | | | |
| 0.1 | 62.18 | **63.39** | 24.58 | 48.37 | 39.02 | 46.50 | 47.34 |
| 0.2 | **62.28** | 63.17 | 27.58 | 47.99 | 39.63 | 62.75 | 50.57 |
| 0.3 | 61.99 | 62.68 | 30.46 | 49.66 | 40.24 | **68.50** | **52.26** |
| 0.4 | 61.96 | 62.08 | 32.61 | 49.20 | 39.63 | 64.50 | 51.66 |
| 0.5 | 61.77 | 61.01 | 33.57 | **50.34** | 39.02 | 63.25 | 51.49 |
| 0.6 | 61.54 | 60.01 | 33.93 | 49.81 | 38.41 | 60.50 | 50.70 |
| 0.7 | 61.40 | 58.85 | 36.81 | 49.05 | 39.02 | 58.00 | 50.52 |
| 0.8 | 61.12 | 57.42 | 38.01 | 49.58 | 40.85 | 60.50 | 51.25 |
| 0.9 | 60.71 | 56.04 | 38.61 | 49.20 | 39.02 | 60.50 | 50.68 |
| 1.0 | 60.50 | 54.33 | **41.01** | 49.73 | **42.68** | 58.00 | 51.04 |

Table 4: Test results of KnOTS (integrated with TIES-Merging) under different linear weighting coefficients, denoted as $\alpha$. During model merging, the same coefficient is applied across different models, consistent with the setting described in Section 4.1.

concatenation step and restricts the merging operation to the $\mathbf{V}^T$ matrices, without altering the singular value matrix. Consequently, the behavior under varying linear weighting coefficients differs from that of other model merging approaches. In the following, we provide both theoretical and empirical analyses.

From the theoretical perspective, we measure the difference between KnOTS and other methods using the Frobenius norm of matrices. To illustrate, we consider a simple case with two matrices prior to concatenation: $\mathbf{W}_1 = \mathbf{U\Sigma V}_1^T$ and $\mathbf{W}_2 = \mathbf{U\Sigma V}_2^T$, where $\alpha$ and $\beta$ represent their respective weighting coefficients. The matrix obtained by KnOTS after merging is given by:

$$\|\mathbf{W}\|_F^2 = \left\|\mathbf{\Sigma}(\alpha\mathbf{V}_1^\top + \beta\mathbf{V}_2^\top)\right\|_F^2 = \alpha^2 \|\mathbf{\Sigma}\|_F^2 + \beta^2 \|\mathbf{\Sigma}\|_F^2 + 2\alpha\beta \operatorname{tr}(\mathbf{\Sigma}^2\mathbf{V}_1^\top\mathbf{V}_2) \qquad (11)$$

Since $\left|\operatorname{tr}(\mathbf{\Sigma}^2\mathbf{V}_1^\top\mathbf{V}_2)\right| \leq \operatorname{tr}(\mathbf{\Sigma}^2) = \|\mathbf{\Sigma}\|_F^2$, it follows that:

$$\left|\,|\alpha| - |\beta|\,\right| \|\mathbf{\Sigma}\|_F \leq \|\mathbf{W}\|_F \leq (|\alpha| + |\beta|) \|\mathbf{\Sigma}\|_F \qquad (12)$$

The Frobenius norm bound of $\mathbf{W}$ is not directly related to that of the original models $\mathbf{W}_1$ and $\mathbf{W}_2$. In contrast, under a conventional weighted approach, the upper bound of the matrix norm is given by:

$$\|\alpha\mathbf{W}_1 + \beta\mathbf{W}_2\|_F \leq |\alpha| \|\mathbf{W}_1\|_F + |\beta| \|\mathbf{W}_2\|_F \qquad (13)$$

Therefore, as the linear weighting coefficients vary, the results of the latter merging approach can change substantially, whereas the outcomes of KnOTS may exhibit less variation.

We also conduct experimental validation using the setup that combines the four models listed in Table 1. Specifically, we examine the performance of KnOTS (integrated with TIES-Merging) as the range of linear weighting coefficients varies within $[0.1, 1.0]$, as described in Section 4.1.

As shown in Table 4, the performance of the KnOTS method does not exhibit substantial variation with different linear weighting coefficients, which contrasts with the trends observed in Figure 4b. Therefore, directly comparing KnOTS with other methods under the same settings of linear weighting coefficients would be unfair. We further speculate that this behavior of KnOTS may stem from its matrix concatenation procedure and the operation that only the $\mathbf{V}$ matrix from the SVD results is merged.

# C    ADDITIONAL EXPERIMENTS

**Wall-clock time experiments.** We test the wall-clock time of different methods under the experimental settings in Table 1, and examine how the merging time of each method varies with the number of models $n$. The experimental results are presented in Table 5. The results show that when $n \geq 7$, the original method requires more than $80GB$ of training memory and reports an OOM error. Moreover, as $n$ increases, its actual runtime grows much faster than that of the optimized version, whereas the runtime of our TSPA method increases approximately linearly with $n$.

**Different random seeds.** In Table 1, we use $42$ as the random seed. To demonstrate the robustness of TSPA, we additionally set five random seeds and conduct experiments, as shown in Table 6. The results show that TSPA is insensitive to the random seed and exhibits good robustness.

**DoRA.** We retrain the DoRA adapters with $rank = 8$ based on the experimental setup in Table 1 and evaluate the performance of the different methods. The experimental results are presented in Table 7.

| Merging Methods | $n$-Values | | | | | | |
|---|---|---|---|---|---|---|---|
| | $n = 2$ | $n = 3$ | $n = 4$ | $n = 5$ | $n = 6$ | $n = 7$ | $n = 8$ |
| Task Arithmetic | 8 s | 8 s | 8 s | 8 s | 8 s | 9 s | 9 s |
| TIES | 8 s | 8 s | 8 s | 9 s | 9 s | 9 s | 9 s |
| DARE+TIES | 8 s | 8 s | 8 s | 9 s | 9 s | 9 s | 9 s |
| KnOTS+TIES | 1 min 23 s | 1 min 57 s | 1 min 43 s | 1 min 45 s | 1 min 44 s | 1 min 45 s | 1 min 49 s |
| EMR-Merging | 8 s | 8 s | 8 s | 9 s | 9 s | 9 s | 11 s |
| PCB-Merging | 8 s | 8 s | 8 s | 9 s | 9 s | 9 s | 9 s |
| TSPA(Attention, original) | 9 min 24 s | 14 min 18 s | 19 min 11 s | 26 min 5 s | 34 min 27 s | **OOM** | **OOM** |
| TSPA(Attention) | 7 min 19 s | 8 min 46 s | 10 min 13 s | 11 min 36 s | 13 min 2 s | 14 min 27 s | 15 min 55 s |
| TSPA(LoRA) | 4 min 2 s | 4 min 22 s | 5 min 16 s | 5 min 55 s | 7 min 3 s | 8 min 9 s | 9 min 26 s |
| TSPA | 10 min 38 s | 13 min 17 s | 16 min 40 s | 18 min 5 s | 20 min 52 s | 23 min 14 s | 25 min 30 s |

Table 5: Test results of wall-clock time.

| Merging Methods | General Capabilities | | Instruction-Following | Math | Code | Safety | Avg. |
|---|---|---|---|---|---|---|---|
| | MMLU | TriviaQA | IFEval | GSM8K | HumanEval | DirectHarm4 | |
| Original | 62.19 | 63.34 | 21.34 | 38.36 | 35.37 | 29.25 | 41.64 |
| Fine-tuned | / | / | 25.78 | 45.19 | 42.68 | 78.25 | / |
| TSPA(seed=0) | 61.75 | 62.39 | 25.90 | **49.81** | 40.24 | 77.50 | 52.93 |
| TSPA(seed=42) | **62.16** | **62.78** | **26.74** | 49.43 | **40.85** | 72.50 | 52.41 |
| TSPA(seed=123) | 61.94 | 62.47 | 26.62 | 49.73 | 39.63 | 76.25 | 52.77 |
| TSPA(seed=2025) | 61.75 | 62.41 | 25.90 | **49.81** | 40.24 | **78.25** | **53.06** |
| TSPA(seed=314159) | 61.94 | 62.47 | 25.42 | 49.73 | 39.63 | **78.25** | 52.91 |

Table 6: Test results of different random seeds.

# D    THE USE OF LARGE LANGUAGE MODELS

We commit to accurately reporting our use of LLMs. Specifically, we use LLMs to polish writing and to assist with the discovery of related work. However, we do not use LLMs to generate the core ideas or to write substantive portions of this paper.

| Merging Methods | General Capabilities | | Instruction-Following | Math | Code | Safety | Avg. |
|---|---|---|---|---|---|---|---|
| | MMLU | TriviaQA | IFEval | GSM8K | HumanEval | DirectHarm4 | |
| Original | 62.19 | 63.34 | 21.34 | 38.36 | 35.37 | 29.25 | 41.64 |
| Fine-tuned | / | / | 25.78 | 45.19 | 42.68 | 78.25 | / |
| TA | 22.95 | 0.01 | 22.66 | 0.00 | 0.00 | 0.00 | 7.60 |
| TIES | 62.41 | **63.63** | 24.94 | 45.26 | 36.59 | 63.75 | 49.43 |
| DARE+TIES | 62.37 | 63.44 | 24.46 | 44.05 | 38.41 | 66.75 | 49.91 |
| EMR-Merging | 26.89 | 0.01 | 13.07 | 0.00 | 0.00 | 0.00 | 6.66 |
| PCB-Merging | **62.46** | 62.85 | 24.46 | 44.81 | 35.98 | 67.25 | 49.64 |
| TSPA(Attention) | 62.04 | 60.11 | **30.10** | **49.05** | **40.24** | **81.25** | **53.80** |
| TSPA(LoRA) | 61.32 | 62.81 | 27.70 | 44.81 | 37.20 | 66.25 | 50.02 |
| TSPA | 62.02 | 62.43 | 26.62 | 47.99 | **40.24** | 82.25 | 53.59 |

Table 7: Test results of different methods for merging DoRA adapters.