# OpenReview forum: "Leveraging Rotation Symmetry for Efficient LoRA Merging in Large Language Models"
_ICLR.cc/2026/Conference — ICLR 2026 Conference Withdrawn Submission_

### Official Review · Reviewer_ggVN · 2025-10-16

**Soundness:** 3
**Presentation:** 3
**Contribution:** 3
**Rating:** 4
**Confidence:** 4

**Summary:**

The paper investigates the problem of parameter interference when merging multiple low-rank adaptations (LoRAs) in large language models, which often leads to performance degradation. It proposes a Two-Stage Parameter Alignment (TSPA) framework that applies rotation symmetry to align parameters in low-rank space, addressing structural incompatibility and reducing computational complexity from quadratic to linear. The core contributions include a novel alignment mechanism and a linear-complexity paradigm. Experimental results on models like Llama-3-8B demonstrate that TSPA outperforms state-of-the-art methods in average performance across NLP tasks, robustness in high-rank scenarios, and retention of fine-grained capabilities such as safety.

**Strengths:**

1. The paper introduces an innovative low-rank space alignment mechanism that preserves LoRA structure while maintaining functional equivalence, solving the problem of vulnerability when merging high-rank LoRAs.

2. The paper discusses safety, which is a perspective that mainstream merging methods ignore and is of great significance.

3. It covers comprehensive evaluations across multiple tasks (e.g., Instruction-Following, Math, Code, Safety), different LoRA ranks (e.g., 8, 16, 32), and models (e.g., Llama-3-8B, Qwen3-8B).

**Weaknesses:**

1. This paper missed several key researchs that are highly related to the motivation. For example, RobustMerge[1] also claims to tackle robustness when merging LoRA. It would be of great benefit to have a comprehensive comparison with it to demonstrate the effectiveness of the proposed method. Other works like KnOTS and LoraHub should also be conducted.

2. The experimental results are not that impressive. The improvements mainly come from safety task.

3. No explicit explanation of O(n) in the introduction. It appears afterwards and the authors should reorganize the order.

4. The figures in Figure 3 are too small to be read. It should be replaced with a clearer one.

5. The method assumes rotation symmetry applies broadly, but no discussion of scenarios where alignment might fail (e.g., non-linear interactions) is included.

**Questions:**

1. It would be better to compare with LoRA-based[1,2] and up-to-date[3,4] merging methods to validate the effectiveness of the proposed method.

2. More experimental results like different types of structure (mllm, etc), different types of lora (dora, etc) are encouraged to further demonstrate the effectiveness of the proposed method.

3. The paper mentions computational efficiency in merging. But I think the complexity mainly comes from the iterative process it introduces in merging. A large number of merging methods like DARE, Ties-merging directly merge multiple models in a post-training paradigm, so the computational efficiency should not be a bottleneck. The author should provide more analysis and experiments, such as quantitative experiments to illustrate the necessity of reducing computational complexity.

4. Despite the proposed TSPA, the technique within each stage like rotation and scaling is not that novel. More explanation and comparison with existing methods should be incorporated.

5. About OOD generalization: How does TSPA scale with the number of LoRAs beyond the tested scenarios?

[1] RobustMerge: Parameter-Efficient Model Merging for MLLMs with Direction Robustness.

[2] LoraHub: Efficient Cross-Task Generalization via Dynamic LoRA Composition.

[3] EMR-Merging: Tuning-Free High-Performance Model Merging.

[4] Parameter Competition Balancing for Model Merging.

---

> ### Author Response · Authors · 2025-11-23
> **Response to Reviewer ggVN (Part 1)**
>
> Thank you for your thoughtful feedback. We provide our responses to each of your points below.
>
> > **Weakness 1: This paper missed several key researchs that are highly related to the motivation. For example, RobustMerge[1] also claims to tackle robustness when merging LoRA. It would be of great benefit to have a comprehensive comparison with it to demonstrate the effectiveness of the proposed method. Other works like KnOTS and LoraHub should also be conducted.**
>
> For RobustMerge, we conduct experiments under both the experimental settings described in the paper and those outlined in the Additional Experiments section above. We find that RobustMerge **fails** to generalize under these conditions because the performance of the merged models is essentially **garbled**. Therefore, we do not report these experimental results.
>
> For the KnOTS method, we provide a detailed analysis in **Appendix B**. However, since KnOTS tends to amplify parameter scaling to some extent and is minimally affected by the weighting coefficients, a direct comparison with other methods would be **unfair**. This is also evident in Table 4, where KnOTS exhibits substantially higher performance on IFEval and GSMK than the fine-tuned models, but this improvement does not reflect the intrinsic effectiveness of the method.
>
> > **Weakness 2: The experimental results are not that impressive. The improvements mainly come from safety task.**
>
> In fact, the performance of our method is quite close to that of individually fine-tuned models, which also represents the **upper bound** for merging performance. At the same time, our method consistently outperforms existing SOTA approaches, such as TIES and DARE, under various experimental settings, demonstrating that TSPA achieves strong performance.
>
> > **Weakness 3: No explicit explanation of O(n) in the introduction. It appears afterwards and the authors should reorganize the order.**
>
> Thank you for your careful observation. $n$ refers to the number of models, and we have updated this clarification in the PDF version of the paper.
>
> > **Weakness 4: The figures in Figure 3 are too small to be read. It should be replaced with a clearer one.**
>
> Thank you for your suggestion. We have updated Figure 3 in the PDF version of the paper to clearer images.
>
> > **Weakness 5: The method assumes rotation symmetry applies broadly, but no discussion of scenarios where alignment might fail (e.g., non-linear interactions) is included.**
>
> In practice, since we have explicit optimization objectives (Equations 7 and 9) and the convergence of the algorithm has been established in [1], TSPA consistently iterates toward and converges on these objectives. On the other hand, due to the sparsity of neural network parameters, we can always identify a series of models that satisfy linear mode connectivity (LMC). This has been confirmed across various experimental settings, and in scenarios where the TA method fails, TSPA is consistently able to achieve successful parameter alignment, resulting in well-performing merged models.
>
> [1] Riemannian adaptive optimization methods

---

> ### Author Response · Authors · 2025-11-23
> **Response to Reviewer ggVN (Part 2)**
>
> > **Question 1: It would be better to compare with LoRA-based[1,2] and up-to-date[3,4] merging methods to validate the effectiveness of the proposed method.**
>
> The methods you mentioned, including RobustMerge, LoraHub, EMR-Merging, and PCB-Merging, are indeed advanced and highly relevant approaches, and we agree that they are valuable for comparison. However, regarding RobustMerge, after running multiple experiments based on the released source code, we find that it does not generalize to the tasks in this paper, and the merged models collapse. Therefore, we do not plan to include RobustMerge as an additional baseline. As for LoraHub, it is not a data-free method. It evaluates the merged model using a small set of examples and essentially tunes the coefficients of the linear combination. For this reason, we believe that adding LoraHub as a baseline would not be appropriate.
>
> In contrast, we include EMR-Merging and PCB-Merging in our experimental setup. We report below the results for merging all four models as well as all three-model combinations. Similarly, in the **Additional Experiments** section above, we also include these two baselines.
>
> **(1) IF+Math+Code+Safety:**
>
> | Model              | MMLU  | TriviaQA | IFEval | GSM8K | HumanEval | DirectHarm4 | Avg.  |
> |--------------------|:-----:|:--------:|:------:|:-----:|:---------:|:-----------:|:-----:|
> | Original           | 62.19 | 63.34    | 21.34  | 38.36 | 35.37     | 29.25       | 41.64 |
> | Fine-tuned         | /     | /        | 28.42  | 43.82 | 41.46     | 89.75       | /     |
> | Concatenate        | 53.72 | 31.10    | 16.09  | 0.53  | 37.80     | 0.00        | 23.21 |
> | Task Arithmetic    | 22.95 | 0.00     | 25.06  | 0.00  | 0.00      | 0.00        | 8.00 |
> | TIES               | 62.21 | 62.67    | 26.02  | 48.37 | 38.41     | 54.25       | 48.66 |
> | DARE+TIES          | 62.31 | **63.16**    | 25.78  | 48.82 | 36.59     | 57.00       | 48.94 |
> | EMR-Merging        | 22.95 | 0.00     | 0.00   | 38.44 | 35.37     | 28.75       | 20.92 |
> | PCB-Merging        | **62.37** | 62.68    | 27.82  | 46.40 | 36.59     | 59.75       | 49.27 |
> | TSPA(Attention)    | 61.99 | 57.82    | **29.62**  | 48.75 | 39.63     | **80.25**       | **53.01** |
> | TSPA(LoRA)         | 61.29 | 62.86    | 28.78  | 45.34 | 37.20     | 64.25       | 49.95 |
> | TSPA               | 62.16 | 62.78    | 26.74  | **49.43** | **40.85**     | 72.50       | 52.41 |
>
> **(2) IF+Math+Code:**
>
> | Model              | MMLU  | TriviaQA | IFEval | GSM8K | HumanEval | Avg.  |
> |--------------------|:-----:|:--------:|:------:|:-----:|:---------:|:-----:|
> | Original           | 62.19 | 63.34    | 21.34  | 38.36 | 35.37     | 44.12 |
> | Fine-tuned         | /     | /        | 28.42  | 43.82 | 41.46     | /     |
> | Task Arithmetic    | 22.95 | 0.00 | 25.30 | 0.00 | 0.00 | 9.65 |
> | TIES               | 62.07 | 63.41    | 24.82  | 48.60 | **40.24**     | **47.83** |
> | DARE+TIES          | 62.18 | 63.09    | 25.66  | 48.07 | 37.20     | 47.24 |
> | EMR-Merging        | 22.95 | 0.00     | 0.00   | 0.00  | 0.00      | 4.59  |
> | PCB-Merging        | **62.23** | **63.58**    | 25.90  | 47.08 | 37.20     | 47.20 |
> | TSPA               | 61.99 | 63.52    | **26.50**  | **49.28** | 35.37     | 47.33 |
>
> **(3) IF+Math+Safety:**
>
> | Model              | MMLU  | TriviaQA | IFEval | GSM8K | DirectHarm4 | Avg.  |
> |--------------------|:-----:|:--------:|:------:|:-----:|:-----------:|:-----:|
> | Original           | 62.19 | 63.34 | 21.34 | 38.36 | 29.25 | 42.90 |
> | Fine-tuned         | /     | /     | 28.42 | 43.82 | 89.75 | /     |
> | Task Arithmetic    | 22.95 | 0.00 | 24.82 | 0.00 | 0.00 | 9.55 |
> | TIES               | 62.32 | 62.30 | 26.26 | **49.05** | 55.50 | 51.09 |
> | DARE+TIES          | 62.21 | 62.97 | **26.74** | 48.75 | 61.75 | 52.48 |
> | EMR-Merging        | 22.95 | 0.00 | 23.14 | 0.00 | 0.00 | 9.22 |
> | PCB-Merging        | **62.42** | 62.38 | 25.90 | 48.60 | 70.75 | 54.01 |
> | TSPA               | 62.40 | **63.40** | 24.46 | 47.23 | **73.25** | **54.15** |
>
> **(4) IF+Code+Safety:**
>
> | Model              | MMLU  | TriviaQA | IFEval | HumanEval | DirectHarm4 | Avg. |
> |--------------------|:-----:|:--------:|:------:|:-----:|:---------:|:-----:|
> | Original           | 62.19 | 63.34 | 21.34 | 35.37 | 29.25 | 42.30 |
> | Fine-tuned         | /     | /     | 28.42 | 41.46 | 89.75 | /     |
> | Task Arithmetic    | 23.32 | 0.00  | 26.74 | 0.00  | 0.00  | 10.01 |
> | TIES               | 61.86 | 62.69 | 26.86 | **39.02** | 65.00 | 51.09 |
> | DARE+TIES          | **62.22** | **62.87** | **27.46** | 38.41 | 65.00 | 51.19 |
> | EMR-Merging        | 26.89 | 0.02  | 0.00  | 0.00  | 0.00  | 5.38  |
> | PCB-Merging        | 62.11 | 62.43 | 26.38 | **39.02** | 66.25 | 51.24 |
> | TSPA               | 62.07 | **62.87** | 26.74 | 37.80 | **84.25** | **54.75** |

---

> ### Author Response · Authors · 2025-11-23
> **Response to Reviewer ggVN (Part 3)**
>
> **(5) Math+Code+Safety:**
>
> | Model              | MMLU  | TriviaQA | GSM8K | HumanEval | DirectHarm4 | Avg.  |
> |--------------------|:-----:|:--------:|:-----:|:---------:|:-----------:|:-----:|
> | Original           | 62.19 | 63.34    | 38.36 | 35.37     | 29.25       | 45.70 |
> | Fine-tuned         | /     | /        | 43.82 | 41.46     | 89.75       | /     |
> | Task Arithmetic    | 22.95 | 0.02     | 0.00  | 0.00      | 0.00        | 4.59  |
> | TIES               | 61.89 | 62.68    | 47.84 | **39.63**     | 49.50       | 52.31 |
> | DARE+TIES          | 62.23 | 62.97    | 47.23 | 39.02     | 50.75       | 52.44 |
> | EMR-Merging        | 22.95 | 0.00     | 0.00  | 0.00      | 0.00        | 4.59  |
> | PCB-Merging        | **62.30** | 62.54    | 47.38 | 36.59     | 63.00       | 54.36 |
> | TSPA               | 61.89 | **63.33**    | **47.99** | 37.20     | **66.50**       | **55.38** |
>
> > **Question 2: More experimental results like different types of structure (mllm, etc), different types of lora (dora, etc) are encouraged to further demonstrate the effectiveness of the proposed method.**
>
> Our work focuses on the merging of LLMs for generative tasks. However, we believe that parameter alignment is a general approach that can also be effective for MLLMs or other LoRA-based variants. These topics are beyond the scope of the present study and will be explored more thoroughly in our future work.
>
> > **Question 3: The paper mentions computational efficiency in merging. But I think the complexity mainly comes from the iterative process it introduces in merging. A large number of merging methods like DARE, Ties-merging directly merge multiple models in a post-training paradigm, so the computational efficiency should not be a bottleneck. The author should provide more analysis and experiments, such as quantitative experiments to illustrate the necessity of reducing computational complexity.**
>
> The issue you raised is indeed critical. In the **Additional Experiments** section above, we add a comparison of wall-clock times across various methods. Our approach reduces the number of original operations from $\frac{n(n−1)}{2}$ to $n$, achieving a $\mathbf{\frac{n−1}{2}}$**x speedup**. As a result, when the number of models $n$ is large, this leads to substantial time savings.
>
> > **Question 4: Despite the proposed TSPA, the technique within each stage like rotation and scaling is not that novel. More explanation and comparison with existing methods should be incorporated.**
>
> In fact, our paper extends rotation symmetry to LoRA merging. While the rotation and scaling operations involved are indeed common practices, our main contribution lies in providing **an end-to-end solution** that achieves parameter alignment through Stiefel manifold optimization. On the other hand, methods such as TSPA and the original Rotation Symmetry method also fall under the parameter alignment methods. Currently, the more stable and effective approaches are typically sparsification-based methods, such as TIES and DARE. Our method, however, offers **an alternative perspective**: when parameter interference is severe, parameter alignment can also effectively mitigate such interference.
>
> > **Question 5: About OOD generalization: How does TSPA scale with the number of LoRAs beyond the tested scenarios?**
>
> Regarding the merging of LLMs for generative tasks, existing methods typically follow the approach used in the DARE paper, testing with three models fine-tuned on Llama-2-13B. Our experiments, involving the merging of four models, demonstrate the limitations of the TA method and show that TSPA can effectively achieve parameter alignment.

---

> > ### Comment · Reviewer_ggVN · 2025-11-24
> >
> > Thank you for your reply. I have several additional questions.
> > 1. Your additional experiment on the wall-clock time corresponds to my question 3, and that your claimed $\mathbf{\frac{n−1}{2}}$x speedup is at the sacrifice of much longer merging time and compare with your original method. So the acceleration is not impressive with much longer merging time compared to existing methods. You mention  "As a result, when the number of models $n$ is large, this leads to substantial time savings", but you do not show the results when $n$ is large, which can not convince me.
> > 2. I don't understand why RobustMerge fails and comparing with KnOTS is unfair. Also, the implementation of PCB-Merging and TA seems questionable as many of them are zero. May I know the reason?
> > 3. You mention "have updated this clarification in the PDF version of the pape", but the changes are not highlighted and are difficult to see the changes.
> > 4. I do not see the update in the manuscript about the additional experimnets, discussion with related works like RobustMerge, PCB-Merging and so on. May I know the reason?
> > 5. Regarding experiments, many of the expected experiments are not conducted including but not limited to Q2 and Q5.

---

> > > ### Author Response · Authors · 2025-12-02
> > > **Official Comment by Authors (Part 1)**
> > >
> > > Thank you for your professional question. We address it from the following perspectives.
> > >
> > > > **Question 1: Your additional experiment on the wall-clock time corresponds to my question 3, and that your claimed $\mathbf{\frac{n−1}{2}}$x speedup is at the sacrifice of much longer merging time and compare with your original method. So the acceleration is not impressive with much longer merging time compared to existing methods. You mention "As a result, when the number of models $n$ is large, this leads to substantial time savings", but you do not show the results when $n$ is large, which can not convince me.**
> > >
> > > We measure the wall-clock time for merging different numbers of models. The results show that when $n \geq 7$, the original method requires more than 80 GB of training memory and reports an OOM error. Moreover, as $n$ increases, its actual runtime grows much faster than that of the optimized version, whereas the runtime of our TSPA method increases approximately linearly with $n$.
> > >
> > > | Method             | $n=2$ | $n=3$ | $n=4$ | $n=5$ | $n=6$ | $n=7$ | $n=8$ |
> > > |--------------------|:-----:|:-----:|:-----:|:-----:|:-----:|:-----:|:-----:|
> > > | Task Arithmetic    | 8 s   | 8 s   | 8 s   | 8 s   | 8 s   | 9 s   | 9 s   |
> > > | TIES               | 8 s   | 8 s   | 8 s   | 9 s   | 9 s   | 9 s   | 9 s   |
> > > | DARE+TIES          | 8 s   | 8 s   | 8 s   | 9 s   | 9 s   | 9 s   | 9 s   |
> > > | KnOTS+TIES         | 1 min 23 s | 1 min 57 s | 1 min 43 s | 1 min 45 s | 1 min 44 s | 1 min 45 s | 1 min 49 s |
> > > | EMR-Merging        | 8 s   | 8 s   | 8 s   | 9 s   | 9 s   | 9 s   | 11 s  |
> > > | PCB-Merging        | 8 s   | 8 s   | 8 s   | 9 s   | 9 s   | 9 s   | 9s    |
> > > | TSPA(Attention, original) | 9 min 24 s | 14 min 18 s | 19 min 11 s | 26 min 5 s | 34 min 27 s | **OOM** | **OOM** |
> > > | TSPA(Attention)    | 7 min 19 s | 8 min 46 s | 10 min 13 s | 11 min 36 s | 13 min 2 s | 14 min 27 s | 15 min 55 s |
> > > | TSPA(LoRA)         | 4 min 2 s | 4 min 22 s | 5 min 16 s | 5 min 55 s | 7 min 3 s | 8 min 9 s | 9 min 26 s   |
> > > | TSPA               | 10 min 38 s | 13 min 17 s | 16 min 40 s | 18 min 5 s | 20 min 52 s | 23 min 14 s | 25 min 30 s |
> > >
> > > > **Question 2: I don't understand why RobustMerge fails and comparing with KnOTS is unfair. Also, the implementation of PCB-Merging and TA seems questionable as many of them are zero. May I know the reason?**
> > >
> > > In fact, our experiments are based on an LLM backbone, whereas the evaluation of RobustMerge is conducted on an MLLM, and RobustMerge requires additional tuning of merging coefficients, which does not play a role in this work. Regarding KnOTS, we discuss it in Appendix B. KnOTS achieves performance on IFEval and GSM8K that far exceeds that of individually fine-tuned models. We conjecture that this stems from the unexplained concatenation operation on the $V$-matrix in the KnOTS method. Furthermore, the final results show that KnOTS exhibits relatively similar performance across different merging coefficients. For TIES, DARE+TIES, and TSPA, the average accuracy **drops by 5.45%, 6.38%, and 1.52%** (absolute differences, not relative), respectively. Therefore, we believe that KnOTS, to some extent, amplifies the magnitude of the task vector, leading to these peculiar results. Nevertheless, we add the KnOTS+TIES results in Table 1.
> > >
> > > We speculate that you are referring to EMR-Merging rather than PCB-Merging, since PCB-Merging performs reasonably well. For TA, we adopt the implementation provided in mergekit, whereas for EMR-Merging we use the original codebase. Thus, the issue does not lie in our implementation. When the merging coefficient $\alpha$ is reduced to 0.2, the outputs of TA and EMR-Merging are no longer garbled, and when $\alpha$ is reduced to 0.1, the merged results become quite satisfactory. Hence, many of the zero outputs they report arise from parameter interference, and reducing $\alpha$ mitigates this issue to some extent. Similarly, reducing the number of merged models $n$ also alleviates the problem.

---

> > > > ### Author Response · Authors · 2025-12-02
> > > > **Official Comment by Authors (Part 2)**
> > > >
> > > > | Method             | MMLU  | TriviaQA | IFEval | GSM8K | HumanEval | DirectHarm4 | Avg.  |
> > > > |--------------------|:-----:|:--------:|:------:|:-----:|:---------:|:-----------:|:-----:|
> > > > | Original           | 62.19 | 63.34    | 21.34  | 38.36 | 35.37     | 29.25       | 41.64 |
> > > > | Fine-tuned         | /     | /        | 28.42  | 43.82 | 41.46     | 89.75       | /     |
> > > > | **$\alpha$ = 1.0** |       |          |        |       |           |             |       |
> > > > | TA                 | 22.95 | 0.00     | 25.06  | 0.00  | 0.00      | 0.00        | 8.00  |
> > > > | EMR-Merging        | 22.95 | 0.00     | 0.00   | 38.44 | 35.37     | 28.75       | 20.92 |
> > > > | **$\alpha$ = 0.2** |       |          |        |       |           |             |       |
> > > > | TA                 | 58.40 | 32.97    | 23.65  | 39.95 | 41.46     | 71.25       | 44.61 |
> > > > | EMR-Merging        | 56.47 | 24.55    | 22.57  | 39.95 | 35.98     | 55.25       | 39.13 |
> > > > | **$\alpha$ = 0.1** |       |          |        |       |           |             |       |
> > > > | TA                 | 61.96 | 61.32    | 32.13  | 48.52 | 40.85     | 77.00       | 53.63 |
> > > > | EMR-Merging        | 61.36 | 59.46    | 32.73  | 50.34 | 40.24     | 79.25       | 53.90 |
> > > >
> > > > > **Question 3: You mention "have updated this clarification in the PDF version of the pape", but the changes are not highlighted and are difficult to see the changes.**
> > > >
> > > > Thank you for your reminder. We mark the revised parts in blue.
> > > >
> > > > > **Question 4: I do not see the update in the manuscript about the additional experimnets, discussion with related works like RobustMerge, PCB-Merging and so on. May I know the reason?**
> > > >
> > > > Thank you for your careful review. We add references and descriptions of EMR-Merging, PCB-Merging, and RobustMerge in the Related Work section.
> > > >
> > > > > **Question 5: Regarding experiments, many of the expected experiments are not conducted including but not limited to Q2 and Q5.**
> > > >
> > > > Your perspective is highly professional. However, since we focus on the LLM backbone, our primary concern is the performance of LLMs on generative tasks. Increasing the number of MLLM experiments would be excessive and is therefore beyond the scope of our discussion. Regarding LoRA variants, although previous work does not include comparisons in this aspect, we consider it valuable to add an ablation study here. We retrain DoRA with $rank=8$ according to the settings in Table 1 and conduct the corresponding experiments. The results are shown in the table below.
> > > >
> > > > **DoRA:**
> > > >
> > > > | Method             | MMLU  | TriviaQA | IFEval | GSM8K | HumanEval | DirectHarm4 | Avg.  |
> > > > |--------------------|:-----:|:--------:|:------:|:-----:|:---------:|:-----------:|:-----:|
> > > > | Original           | 62.19 | 63.34    | 21.34  | 38.36 | 35.37     | 29.25       | 41.64 |
> > > > | Fine-tuned         |   /   |    /     | 25.78  | 45.19 | 42.68     | 78.25       |   /   |
> > > > | TA                 | 22.95 |  0.01    | 22.66  | 0.00  | 0.00      | 0.00        |  7.60 |
> > > > | TIES               | 62.41 | 63.63    | 24.94  | 45.26 | 36.59     | 63.75       | 49.43 |
> > > > | DARE+TIES          | 62.37 | 63.44    | 24.46  | 44.05 | 38.41     | 66.75       | 49.91 |
> > > > | EMR-Merging        | 26.89 |  0.01    | 13.07  | 0.00  | 0.00      | 0.00        |  6.66 |
> > > > | PCB-Merging        | 62.46 | 62.85    | 24.46  | 44.81 | 35.98     | 67.25       | 49.64 |
> > > > | TSPA(Attention)    | 62.04 | 60.11    | 30.10  | 49.05 | 40.24     | 81.25       | 53.80 |
> > > > | TSPA(LoRA)         | 61.32 | 62.81    | 27.70  | 44.81 | 37.20     | 66.25       | 50.02 |
> > > > | TSPA               | 62.02 | 62.43    | 26.62  | 47.99 | 40.24     | 82.25       | 53.59 |

---

### Official Review · Reviewer_L1Rc · 2025-10-16

**Soundness:** 2
**Presentation:** 2
**Contribution:** 3
**Rating:** 4
**Confidence:** 3

**Summary:**

This paper addresses the challenge of parameter interference in lora merging, which often leads to significant performance degradation. The authors identify that existing rotation alignment methods are inapplicable to lora merging due to structural incompatibility and quadratic complexity. To addaress this, the authors introduces the two-stage parameter alignment (TSPA) framework. The main contributions include an alignment mechanism that operates within the lora low-rank space to preserve its structure, and a `comparison with an average model' paradigm that reduces complexity to linear. Empirical results on Llama-3-8B demonstrate that TSPA outperforms SOTA methods, particularly in high-interference settings like high-rank lora merging.

**Strengths:**

- The primary strength of this paper is the adaptation of rotation symmetry for the specific constraints of lora merging. The solution to the structural incompatibility problem by operating directly on the low-rank matrices is effective.

- The `comparison-to-average-model' strategy is interesting, which reduces the computational complexity from quadratic to linear.

**Weaknesses:**

- The paper focuses on the reduction in asymptotic complexity. However, from a practical stand point, some analysis of the wall-clock time and computational overhead of the iterative optimization process, especially in comparison to the non-iterative baselines, would be beneficial.

- The paper uses a simple linear average of the models as the alignment anchor. A discussion on the method's sensitivity to the quality of this initial anchor point could be beneficial, particularly in cases of extreme initial parameter interference.

- An analysis of the model's sensitivity to the hyperparameters is needed to further understand the paper's claims of robustness.

**Questions:**

- Could you provide some comparison of the wall-clock time required for merging using TSPA versus the TIES-Merging and DARE baselines under the experimental settings reported in Table 1?

- The two alignment stages are performed sequentially. What is the rationale for this design choice over a joint optimization of the two objectives?

- How does the alignment process perform if the initial 'average model' target is of very low quality (as in the `Task Arithmetic' case)? Does the optimization procedure reliably converge to a good solution regardless of the anchor point's quality?

---

> ### Author Response · Authors · 2025-11-23
> **Response to Reviewer L1Rc**
>
> Thank you for your valuable perspective and careful reading. We reply to your questions through the following explanations.
>
> > **Weakness 1: The paper focuses on the reduction in asymptotic complexity. However, from a practical stand point, some analysis of the wall-clock time and computational overhead of the iterative optimization process, especially in comparison to the non-iterative baselines, would be beneficial.**
>
> The point you raised is very important, and we are **sorry** for not clarifying it in the paper. We report the wall-clock time of different methods in the **Additional Experiments** section above. Since TSPA requires training, its processing time is longer, but it is still acceptable compared with the time required for fine-tuning.
>
> > **Weakness 2: The paper uses a simple linear average of the models as the alignment anchor. A discussion on the method's sensitivity to the quality of this initial anchor point could be beneficial, particularly in cases of extreme initial parameter interference.**
>
> You are correct that our method treats the parameters produced by TA as an initial anchor point. However, this approach indeed suffers from severe parameter interference, as evidenced by the experiments reported in the paper. In the **Additional Experiments** section above, we further examine the case where the merging coefficient is set to $\frac{1}{n}$. We find that parameter interference becomes more severe as the merging coefficient grows and as the number of models $n$ increases. When merging three models with a coefficient of 0.33, TA still performs reasonably well. However, increasing either of these values causes the merged model to collapse, producing garbled outputs. As a result, the merged parameters no longer satisfy the property of linear mode connectivity (LMC), which means the linearly combined parameters fail to stay in the vicinity of any local optimum.
>
> In contrast, our method consistently uses the TA result only as an initial anchor point, and then optimizes Equations (7) and (9) to identify a set of models that satisfy LMC. As shown by the experimental results in the paper, the final merged models achieve strong overall performance.
>
> > **Weakness 3: An analysis of the model's sensitivity to the hyperparameters is needed to further understand the paper's claims of robustness.**
>
> We additionally analyze the results under different random seeds in the **Additional Experiments** section above, and the experiments demonstrate that our method exhibits good robustness.
>
> > **Question 1: Could you provide some comparison of the wall-clock time required for merging using TSPA versus the TIES-Merging and DARE baselines under the experimental settings reported in Table 1?**
>
> The point you raised is very important. In the **Additional Experiments** section above, we analyze the wall-clock time of different methods.
>
> > **Question 2: The two alignment stages are performed sequentially. What is the rationale for this design choice over a joint optimization of the two objectives?**
>
> Your observation is indeed very insightful. In our experiments, we find that the loss optimized during the "Attention Alignment" stage is much larger, while the loss optimized during the "LoRA Alignment" stage is relatively small. On the one hand, it is difficult for these two stages to maintain a synchronized training pace, which imposes stricter requirements on the hyperparameters during training. On the other hand, a joint optimization of the two objectives demands significantly more GPU memory in practice. Therefore, given the actual training constraints and the observed performance, we perform the two stages sequentially.
>
> > **Question 3: How does the alignment process perform if the initial 'average model' target is of very low quality (as in the `Task Arithmetic' case)? Does the optimization procedure reliably converge to a good solution regardless of the anchor point's quality?**
>
> The issue you raised is indeed critical. In fact, our method uses the weights obtained from Task Arithmetic (TA) as the initial target and continuously optimizes the individual model weights used in the TA computation. As a result, the initial anchor points we employ are suboptimal. However, after parameter alignment, the models’ parameters satisfy LMC property, and the performance of their linear combinations is significantly improved. Our experiments further demonstrate the effectiveness of TSPA.
>
> In addition, our optimizations of Equations 7 and 9 are based on existing Stiefel manifold optimization algorithms, which can be combined with gradient descent methods. The convergence proofs for various gradient descent algorithms are provided in [1].
>
> [1] Riemannian adaptive optimization methods

---

### Official Review · Reviewer_QzJ7 · 2025-10-25

**Soundness:** 3
**Presentation:** 3
**Contribution:** 3
**Rating:** 4
**Confidence:** 4

**Summary:**

This paper introduces a novel framework called TSPA to solve a critical problem in LLMs: efficiently merging multiple LoRAs without catastrophic performance degradation. Merging LoRAs is essential for deployment efficiency, but it often fails due to parameter interference, where conflicts between the adapters' parameters sharply decrease the merged model's capabilities.

**Strengths:**

- Solves Critical, Previously Unsolved Bottlenecks: The paper successfully addresses two nearly insurmountable obstacles that previously hindered the use of rotation alignment for LoRA merging. It introduces a novel mechanism to perform alignment within the LoRA low-rank space, solving the structural incompatibility issue. Simultaneously, it pioneers an alignment paradigm based on comparison with an average model, which reduces computational complexity from a quadratic growth rate to a scalable linear growth rate.
- Demonstrated Robustness in High-Interference Scenarios: The TSPA framework shows significantly greater robustness than existing SOTA methods, especially in high-interference scenarios. In experiments with high-rank adapters (specifically, a rank of 32), where SOTA methods experience performance collapse, TSPA maintains robust and outstanding results, proving its superiority in handling complex parameter conflicts.
- Achieves a Balanced and Comprehensive Performance: The two-stage design, which combines macro-functional alignment (Attention Alignment) and micro-parameter alignment (LoRA Alignment), achieves an optimal balance between task-specific capabilities and general knowledge. Ablation studies confirm that this integrated approach is more comprehensive and powerful than using either alignment stage alone.
- Superior Retention of Fine-Grained Functionality: Beyond just improving average task performance, TSPA demonstrates a unique advantage in preserving delicate and fragile model functionalities. In all experiments, TSPA consistently outperformed other methods in safety capability evaluations , indicating its alignment strategy is more effective at protecting fine-grained parameter structures vulnerable during merging.

**Weaknesses:**

- `Limited Comparison`: The study restricts its comparison to limited TSPA baselines, omitting a comprehensive analysis against more recent work [1]. Furthermore, it lacks exploration of other LoRA variants [2-4], potentially limiting the method's applicability to the broader PEFT landscape.
- `Limited Analysis`: The paper's approach of optimizing on the Stiefel manifold to solve complex and constrained problems is heuristic. A deeper investigation into theoretical bounds and performance preservation guarantees is warranted.
- `Limited Results`: Despite claiming efficient time complexity, the paper presents no supporting experimental results. This omission hinders a clear assessment of its performance-efficiency trade-offs against competing methods.
- `Limited Open-sourcing`: The open-source contribution is confined to training logs without the corresponding test logs, which offers limited value for verification. A more comprehensive and transparent release is expected.

[1] SafeMERGE: Preserving Safety Alignment in Fine-Tuned Large Language Models via Selective Layer-Wise Model Merging

[2] When MOE Meets LLMs: Parameter Efficient Fine-tuning for Multi-task Medical Applications

[3] HydraLoRA: An Asymmetric LoRA Architecture for Efficient Fine-Tuning

[4] CoLA: Collaborative Low-Rank Adaptation

**Questions:**

See Weaknesses.

---

> ### Author Response · Authors · 2025-11-23
> **Response to Reviewer QzJ7**
>
> Thank you for your review. We respond to your concerns from the following perspectives.
>
> > **Weakness 1: Limited Comparison: The study restricts its comparison to limited TSPA baselines, omitting a comprehensive analysis against more recent work [1]. Furthermore, it lacks exploration of other LoRA variants [2-4], potentially limiting the method's applicability to the broader PEFT landscape.**
>
> This paper proposes a merging method and an optimization framework. The works you mentioned [1–4] are beyond the scope of our discussion. However, we believe that TSPA represents a generalized LoRA merging framework and is likely to be effective for merging other LoRA variants as well.
>
> > **Weakness 2: Limited Analysis: The paper's approach of optimizing on the Stiefel manifold to solve complex and constrained problems is heuristic. A deeper investigation into theoretical bounds and performance preservation guarantees is warranted.**
>
> In fact, the convergence of optimization on the Stiefel manifold has been formally established in the paper [1]. This is not a heuristic approach, but rather integrates contemporary gradient descent techniques and represents a widely applicable method for manifold optimization. For the Amsgrad algorithm that we use, the proof is as follows:
>
> >> **Theorem 5 (Convergence of Amsgrad).** Let $(f_t)$ be a family of differentiable, convex functions from $\mathbb{R}^n$ to $\mathbb{R}$. Let $(x_t)$ and $(v_t)$ be the sequences obtained from Algorithm 1b, $\alpha_t=\alpha/\sqrt{t}$, $\beta_1=\beta_{11}$, $\beta_{1t}\leq\beta_1$ for all $t\in [T]$ and $\gamma =\beta_1/\sqrt{\beta_2} <1$. Assume that each $X_i\subset\mathbb{R}$ has a diameter bounded by $D_\infty$ and that for all $1\leq i\leq n$, $t\in [T]$ and $x\in X$, $\Vert(\mathrm{grad} f_t(x))\Vert_{\infty}\leq G_\infty$. For $(x_t)$ generated using the Amsgrad (Algorithm 1b), we have the following bound on the regret
> $$
> R\_T \leq \dfrac{\sqrt{T} D\_\infty^2}{2\alpha(1-\beta\_1)}\sum\_{i=1}^n \sqrt{\hat{v}\_T^i} + \dfrac{D\_\infty^2}{2(1-\beta\_1)}\sum\_{i=1}^n\sum\_{t=1}^T\beta\_{1t}\dfrac{\sqrt{\hat{v}\_t^i}}{\alpha\_t}+\\
> \dfrac{\alpha\sqrt{1+\log T}}{(1-\beta\_1)^2(1-\gamma)\sqrt{1-\beta\_2}}\sum\_{i=1}^n\sqrt{\sum\_{t=1}^T (g^i\_t)^2}
> $$
>
> [1] Riemannian adaptive optimization methods
>
> > **Weakness 3: Limited Results: Despite claiming efficient time complexity, the paper presents no supporting experimental results. This omission hinders a clear assessment of its performance-efficiency trade-offs against competing methods.**
>
> What you mentioned is indeed very important. In the **Additional Experiments** section above, we present the actual runtime both before and after our optimization. The observed runtimes are also consistent with our analysis, showing a speedup of approximately $\mathbf{\frac{n−1}{2}=1.5}$ **times**.
>
> > **Weakness 4: Limited Open-sourcing: The open-source contribution is confined to training logs without the corresponding test logs, which offers limited value for verification. A more comprehensive and transparent release is expected.**
>
> In the **Supplementary Material** of our paper, we provide the code, training logs, and environment configurations. If the paper is accepted, we will release the code and models on GitHub to ensure reproducibility and enable end-to-end replication. Therefore, the files and instructions we provide are comprehensive and transparent, and we hope to make further contributions to the open-source community.

---

> > ### Comment · Reviewer_QzJ7 · 2025-11-26
> >
> > Thank you for the response, especially the proof regarding the Amsgrad algorithm. However, the authors’ perfunctory attitude is puzzling. The entire paper centers on the LoRA method, yet existing LoRA variants have already achieved widely recognized progress. Further exploration of these variants would help verify the generality of the proposed method and uncover deeper patterns—this is the very purpose of scientific research.

---

> ### Author Response · Authors · 2025-11-26
> **Response to Comment from Reviewer QzJ7**
>
> Thank you for your reply.
> ### ***If we have offended or overlooked you, we sincerely apologize.***
> We sincerely appreciate your constructive comments. We have carefully addressed your concerns as follows.
>
> **1.Regarding LoRA variants**
>
> For LoRA variants, we adopt the widely used **DoRA** [1] method. We retrain the corresponding models, and conduct the experiments using the settings in Table 1.
>
> | Model              | MMLU  | TriviaQA | IFEval | GSM8K | HumanEval | DirectHarm4 | Avg.  |
> |--------------------|:-----:|:--------:|:------:|:-----:|:---------:|:-----------:|:-----:|
> | Original           | 62.19 | 63.34    | 21.34  | 38.36 | 35.37     | 29.25       | 41.64 |
> | Fine-tuned         |   /   |    /     | 25.78  | 45.19 | 42.68     | 78.25       |   /   |
> | TA                 | 22.95 |  0.01    | 22.66  | 0.00  | 0.00      | 0.00        |  7.60 |
> | TIES               | 62.41 | **63.63**    | 24.94  | 45.26 | 36.59     | 63.75       | 49.43 |
> | DARE+TIES          | 62.37 | 63.44    | 24.46  | 44.05 | 38.41     | 66.75       | 49.91 |
> | EMR-Merging        | 26.89 |  0.01    | 13.07  | 0.00  | 0.00      | 0.00        |  6.66 |
> | PCB-Merging        | **62.46** | 62.85    | 24.46  | 44.81 | 35.98     | 67.25       | 49.64 |
> | TSPA(Attention)    | 62.04 | 60.11    | **30.10**  | **49.05** | **40.24**     | 81.25       | **53.80** |
> | TSPA(LoRA)         | 61.32 | 62.81    | 27.70  | 44.81 | 37.20     | 66.25       | 50.02 |
> | TSPA               | 62.02 | 62.43    | 26.62  | 47.99 | **40.24**     | **82.25**       | 53.59 |
>
> As for the other three LoRA variants you mentioned, **we sincerely believe they fall outside the scope of this work**. Recent model-merging research, such as KnOTS [2], similarly does not incorporate these variants. Moreover, merging LoRA variants typically requires additional algorithmic design, which is not the focus of our study.
>
> **2.Regarding SafeMERGE**
>
> SafeMERGE is a method designed to align a fine-tuned model with a safety-domain model. Therefore, it is not a multi-model merging method. SafeMERGE does not mitigate parameter interference on its own and requires combination with other techniques.
> We integrate SafeMERGE with TIES and conduct experiments following the settings in Table 1. The results are shown below.
>
> | Model              | MMLU  | TriviaQA | IFEval | GSM8K | HumanEval | DirectHarm4 | Avg.  |
> |--------------------|:-----:|:--------:|:------:|:-----:|:---------:|:-----------:|:-----:|
> | Original           | 62.19 | 63.34    | 21.34  | 38.36 | 35.37     | 29.25       | 41.64 |
> | Fine-tuned         | /     | /        | 28.42  | 43.82 | 41.46     | 89.75       | /     |
> | TA                 | 22.95 | 0.00     | 25.06  | 0.00  | 0.00      | 0.00        | 8.00  |
> | TIES               | 62.21 | 62.67    | 26.02  | 48.37 | 38.41     | 54.25       | 48.66 |
> | DARE+TIES          | 62.31 | **63.16**    | 25.78  | 48.82 | 36.59     | 57.00       | 48.94 |
> | EMR-Merging        | 22.95 | 0.00     | 0.00   | 38.44 | 35.37     | 28.75       | 20.92 |
> | PCB-Merging        | **62.37** | 62.68    | **27.82**  | 46.40 | 36.59     | 59.75       | 49.27 |
> | **SafeMERGE+TIES** | 61.91 | 63.09    | 26.86  | 47.54 | 37.80     | 47.75       | 47.49 |
> | TSPA               | 62.16 | 62.78    | 26.74  | **49.43** | **40.85**     | **72.50**       | **52.41** |
>
> [1] DoRA: Weight-Decomposed Low-Rank Adaptation
>
> [2] Model merging with svd to tie the knots
>
> &nbsp;
> ### ***We truly appreciate your constructive feedback. If you have any additional questions or concerns, we would welcome the opportunity to address them and will respond promptly.***

---

> > ### Comment · Reviewer_QzJ7 · 2025-11-26
> >
> > I sincerely thank the author for such a timely response and experiments. First, there is no need to be overly sensitive; scientific research is objective, not subjective. Second, SafeMERGE itself is not limited to the security domain—it is a general-purpose method. The safety concept behind SafeMERGE is similar to the issue of parameter interference during parameter merging, which does not prevent its application in other fields. This is analogous to the well-known MOELoRA [1], which is not limited to the medical domain.
> >
> > Finally, I do not believe that LoRA variants are unsuitable for the field of parameter fusion. Indeed, directly adding standard LoRA to pretrained weights can lead to parameter interference. However, representative evolved LoRA variants [1–3] generally only modify the number of A or B matrices, without requiring additional algorithmic design. It is worth acknowledging that there may not have been (or may have been) experiments on other LoRA variants in the past, but this does not preclude further exploration—for example, dynamically adding different B matrices to pretrained weights, which could also introduce a MoE-like structure.
> >
> > I will maintain my score because I have not seen evidence of further exploration or particularly interesting insights.
> >
> > [1] When MOE Meets LLMs: Parameter Efficient Fine-tuning for Multi-task Medical Applications
> > [2] HydraLoRA: An Asymmetric LoRA Architecture for Efficient Fine-Tuning
> > [3] CoLA: Collaborative Low-Rank Adaptation

---

> > > ### Author Response · Authors · 2025-11-26
> > > **Official Comment by Authors**
> > >
> > > Thank you for your reply.
> > >
> > > Regarding SafeMERGE, we have already conducted the corresponding experiments and provided explanations, and your perspective **does not** conflict with ours.
> > >
> > > We would like to clarify our work is not about *"parameter fusion,"* but about **model merging**. We believe this distinction is clear. The Mixture-of-Experts (MoE) approach is not directly related to our method, and prior work on model merging has not performed such comparisons. Our comparison protocol follows the standard practice in this line of research.
> > >
> > > ### ***We sincerely hope that your comments can be grounded in our paper and offer constructive insights relevant to this field, rather than personal subjective opinions.***

---

> > > > ### Comment · Reviewer_QzJ7 · 2025-11-26
> > > >
> > > > Thank you for your response. First, there’s no need to get overly emotional—emotion does not necessarily equate to truth. Second, model merging/fusion refers to integrating the capabilities of multiple models into a single model so that it can handle multiple tasks simultaneously without the need to load multiple LoRAs separately. In other words, it is the fusion of multiple LoRAs. Since LoRA itself is just a set of parameters, focusing on "parameter fusion" misses the point.
> > > >
> > > > Finally, the paper does not offer many notable contributions. It might be more productive to consider how further exploration of LoRA variants evolved from MoE could lead to interesting insights. The three representative MoE-based LoRA variants presented are not complex to implement, and no rule or convention requires experiments to follow the exact trajectory of previous work—a constraint that would hinder the discovery of more interesting phenomena.

---

> > > > > ### Author Response · Authors · 2025-11-26
> > > > > **Official Comment by Authors**
> > > > >
> > > > > Thank you for your reply.
> > > > >
> > > > > First, our response **does not** convey any "emotional" content. We **merely** highlight certain text in bold. We **kindly** ask that you not be overly sensitive. We **hope** you can focus on the actual work presented in this paper rather than infer our emotions from the wording.
> > > > >
> > > > > Second, you appear to have a substantial misunderstanding of our work. Our paper concentrates on multi-model merging of LoRA adapters, whereas the three MoE papers you mentioned are **entirely unrelated**. We **agree** that "further exploration of LoRA variants evolved from MoE could lead to interesting insights," which is indeed a fascinating direction. We have conducted some preliminary investigation in this area and hope to explore it further in our future work. However, this line of research **is completely outside the scope of** the current paper.
> > > > >
> > > > > Finally, if there remain any misunderstandings regarding model merging, we encourage you to consult the *Related Work* section of our paper, as well as the following comprehensive surveys [1][2]. We look forward to constructive exchanges with you in this field.
> > > > >
> > > > > ### ***In conclusion, our response is objective and rational. We fully respect you and the other reviewers, and we sincerely hope that any valuable comments you may have will contribute to the improvement of our work.***
> > > > >
> > > > > &nbsp;
> > > > >
> > > > > [1] Merge, Ensemble, and Cooperate! A Survey on Collaborative Strategies in the Era of Large Language Models
> > > > >
> > > > > [2] Scaling Intelligence Through Model Merging: A Comprehensive Survey

---

> > > > > > ### Comment · Reviewer_QzJ7 · 2025-11-26
> > > > > >
> > > > > > Thank you very much for the author’s prompt response. First, subjective statements do not necessarily reflect objective facts; I trust that the other reviewers, ACs, and PCs will make their own accurate assessments. Second, the author seems fixated on past work and reluctant to attempt further exploration, even though the cost of such attempts would be low. Instead, the author questions the reviewer’s expertise. Scientific research is inherently a process of divergent thinking, and interesting findings often speak louder than arguments. Finally, I would like to explain why I insist on keeping my confidence score at 5: there were only a little over 200 minutes between my comment and the author’s reply, yet the author claims to have completed a substantial amount of experimentation (DoRA and SafeMERGE). I remain skeptical about this.

---

> > > > > > > ### Comment · Reviewer_XMCd · 2025-11-26
> > > > > > >
> > > > > > > I appreciate the discussion between the author and the reviewer, but I would like to post my opinions here.
> > > > > > >
> > > > > > > - First, SafeMerge adopts model merging to address safety issues in LLMs, while this paper aims to improve the merging performance among multiple models. They are in different scopes. SafeMerge mainly focuses on two models, $W_f$ and $W_s$, and it is not easy to adapt to multiple models. At least the SafeMerge paper does not **explicitly** claim this point. If the reviewer argues that it is a general-purpose method, it would be better to provide evidence.
> > > > > > > - Second, I may miss something, but as far as I know, there is **no** paper on model merging that shows the results on LoRA **variants**. This implies two points. First, experiments on classical LoRA are a common practice in this field. Second, the main body of this field agrees that the results of classical LoRA are sufficient to evaluate the effectiveness of the method. Again, if the reviewer argues it is necessary and important to evaluate the performance under other LoRA variants, some evidence would be appreciated.
> > > > > > > - Finally, my belief is, scientific research is not the same as products. For research, what we focus on is the novelty of the method, the contribution to the field, and the delivery of a prototype. There are hundreds of PEFT methods, thousands of merging methods, and tons of LLMs. Do we need to evaluate the method on all the combinations of all those settings? It is impossible. More importantly, the meaning of research is to see the potential, rather than pursue perfection.
> > > > > > >
> > > > > > > I believe such a discussion would help this paper and enhance its impact. I am glad to see any thoughts from the author, reviewer, and others.

---

> > > > > > > > ### Comment · Reviewer_QzJ7 · 2025-11-27
> > > > > > > >
> > > > > > > > Thank you for the discussion. I would like to clarify the following points:
> > > > > > > >
> > > > > > > > - The SafeMerge method is applicable, though it may require some familiarity with the approach. In addition, the authors did provide experiments (and if those experiments are indeed faithful to real settings, then I apologize for my earlier doubts).
> > > > > > > >
> > > > > > > > - The idea of MoE can be understood as using different LoRAs or matrices to handle different tasks. A straightforward way of model fusion treats MoE-based LoRA variants as a single LoRA. The paper’s key observation is that models A and B, without rotation, fall into suboptimal regions. The two fundamental contributions of the proposed TSPA method are: (i) rotation alignment in low-rank space (structural compatibility), and (ii) alignment with linear complexity based on the averaged model (efficiency). The inherent properties of MoE happen to reinforce both requirements.
> > > > > > > >
> > > > > > > > - - First, compared with naive LoRA, MoE-based LoRA variants exhibit more severe geometric heterogeneity because the experts have different training objectives, making each expert’s LoRA space inherently heterogeneous. Even when applying LoRA to the same task, the incremental matrices of different experts are distributed along different geometric directions. This is essentially consistent with the paper’s description that “as illustrated in Figure 1, models A, B, and C are situated near distinct local optima,” except that in MoE this “dispersion” is structural rather than incidental, thus making the need for alignment even stronger.
> > > > > > > >
> > > > > > > >  - - Second, from the perspective of tasks, MoE’s dynamic routing via gating amplifies the interference caused by LoRA misalignment. When merging multiple MoE-LoRAs for multi-task fusion, such conflicts multiply. Moreover, because MoE inherently has a “modular structure,” the motivation for rotational alignment is even stronger. TSPA’s low-rank rotation emphasizes aligning the geometric directions of the incremental matrices (A and B) so that they are additive within the same symmetric space, and MoE experts deviate even further internally, which further increases the need for such rotation alignment.
> > > > > > > >
> > > > > > > > - - Finally, TSPA’s average-alignment philosophy is naturally consistent with the MoE gating-based aggregation mechanism: aligning LoRA-aligned experts toward an “average expert direction” is mathematically congruent with MoE’s design philosophy (ensemble, balance, weighted fusion).
> > > > > > > >
> > > > > > > > - I agree that no method is perfect. However, I did not argue that the experiments must be based on all types of PEFT methods; rather, I emphasized the three representative MoE-based methods. Moreover, reviewer ggVN also raised this concern, and perhaps he has additional thoughts on the issue.

---

> > > > > > > > > ### Author Response · Authors · 2025-11-27
> > > > > > > > > **Official Comment by Authors**
> > > > > > > > >
> > > > > > > > > Thank you for your reply.
> > > > > > > > >
> > > > > > > > > I hope that you will continue to uphold an appropriate sense of academic rigor and focus your attention on the present work.
> > > > > > > > >
> > > > > > > > > Your statement that "if those experiments are indeed faithful to real settings, then I apologize for my earlier doubts" **still reflects a strong subjective bias**. **If you have any concrete evidence** suggesting that our experimental results are not faithful or not representative of real scenarios, **please provide specific comments** rather than making unfounded assumptions or speculative allegations.
> > > > > > > > >
> > > > > > > > > Likewise, should the paper be accepted, we will release all trained LoRAs, all training logs, and all code for training, merging, and evaluation on GitHub. This is something I can guarantee.
> > > > > > > > >
> > > > > > > > > In addition, your remark that "TSPA’s average-alignment philosophy is naturally consistent with the MoE gating-based aggregation mechanism: aligning LoRA-aligned experts toward an “average expert direction” is mathematically congruent with MoE’s design philosophy (ensemble, balance, weighted fusion)" **reveals a serious conceptual misunderstanding**. This statement is **entirely incorrect and subjective**. I encourage you to continue reading relevant work in this field to further develop your understanding.
> > > > > > > > >
> > > > > > > > > ### ***Finally, if you have seen our previous questions directed to you, please respond to them one by one.***

---

> > > > > > > > > > ### Comment · Reviewer_QzJ7 · 2025-11-27
> > > > > > > > > >
> > > > > > > > > > I really cannot understand why the author claims that there is nothing wrong with the code yet does not provide the relevant code for further review, and instead keeps making promises. I am puzzled by the author’s counter-question, as I have already provided my answer. In any case, the final decision should be left to the ACs/PCs, and any meaningless comments are unnecessary.

---

> > > > > > > > > > > ### Author Response · Authors · 2025-11-27
> > > > > > > > > > > **Official Comment by Authors**
> > > > > > > > > > >
> > > > > > > > > > > Thank you for your reply.
> > > > > > > > > > >
> > > > > > > > > > > Indeed, any meaningless comments are unnecessary. Moreover, you owe us an apology for **your reckless, rude, and malicious behavior**, and a sincere apology to the readers who have seen this series of comments.
> > > > > > > > > > >
> > > > > > > > > > > &nbsp;
> > > > > > > > > > > ### ***We trust that the ACs, SACs, and PCs who have seen these comments will make the appropriate judgment.***

---

> > > > > > > > > > > > ### Comment · Reviewer_QzJ7 · 2025-11-27
> > > > > > > > > > > >
> > > > > > > > > > > > I trust the judgment of the community.

---

> > > > > > > ### Author Response · Authors · 2025-11-26
> > > > > > > **Official Comment by Authors**
> > > > > > >
> > > > > > > Thank you for your reply.
> > > > > > >
> > > > > > > First, I **agree** with your statement that “subjective statements do not necessarily reflect objective facts.” For the same reason, I **kindly** ask that you refrain from introducing subjective biases into an academic discussion.
> > > > > > >
> > > > > > > Second, our work is built upon research from recent years. Therefore, your claim that “the author seems fixated on past work and reluctant to attempt further exploration” is **entirely incorrect**. Moreover, your comment that “even though the cost of such attempts would be low” reflects a strong subjective bias. How was this conclusion reached?
> > > > > > >
> > > > > > > Third, we would appreciate it if you could focus on the content of this work, which we have emphasized repeatedly. Your statement that “Scientific research is inherently a process of divergent thinking, and interesting findings often speak louder than arguments” is vague and does not provide any constructive suggestions for improving our paper.
> > > > > > >
> > > > > > > Fourth, we would like to remind you that your confidence score was not only 5, but was increased from 4 to 5 after your first-round response. We are curious about the reasoning behind this change.
> > > > > > >
> > > > > > > Fifth, you repeatedly described our prior responses as “emotional.” Do you still maintain that view?
> > > > > > >
> > > > > > > Sixth, it appears that you may have overlooked the experiments presented in the “Additional Experiments” section above. Our experiments are still being expanded, so your statement that “the author claims to have completed a substantial amount of experimentation (DoRA and SafeMERGE)” does not reflect the objective facts.
> > > > > > >
> > > > > > > ### ***Lastly, we would like to emphasize that we have great respect for all reviewers, ACs, SACs, and PCs involved in evaluating this work. We believe that anyone who carefully reads the paper as well as our discussion with Reviewer QzJ7 will form an informed and accurate judgment.***

---

> > > > > > > > ### Comment · Reviewer_QzJ7 · 2025-11-26
> > > > > > > >
> > > > > > > > Thank you for the author’s response. First, an explanation regarding the insistence on confidence has already been provided. Second, I have emphasized the new experiments on DoRA and SafeMERGE, which took just over 200 minutes and raise some concerns. Finally, let’s leave the final decision to the ACs/PCs.

---

> > > > > > > > > ### Author Response · Authors · 2025-11-26
> > > > > > > > > **Official Comment by Authors**
> > > > > > > > >
> > > > > > > > > Thank you for your reply.
> > > > > > > > >
> > > > > > > > > It seems that your focus is not on the contribution of this work itself, but rather on trying to find any possible flaw, and you are particularly concerned about why we were able to complete the above experiments in 200 minutes. We provide an explanation here.
> > > > > > > > >
> > > > > > > > > Regarding the DoRA experiments, you can search this page for "DoRA." Reviewer ggVN has already mentioned experiments on other LoRA variants, and we have indeed spent a long time training DoRA (rank = 8) with LLaMA-Factory, as well as conducting the related tests.
> > > > > > > > >
> > > > > > > > > **This is your oversight—you raise doubts without any verification, which reflects a strong subjective bias.**
> > > > > > > > >
> > > > > > > > > As for the SafeMERGE+TIES experiments, in fact, we only add this one additional method, which took less than one hour. The results for the other methods can be found in our "Response to Reviewer ggVN (Part 2)."
> > > > > > > > >
> > > > > > > > > &nbsp;
> > > > > > > > > ### ***Therefore, we would like to inform the respected ACs, SACs, and PCs that Reviewer QzJ7’s comments contain significant subjective bias and do not provide any meaningful or constructive feedback for this work.***

---

> > > > > > > > > > ### Comment · Reviewer_QzJ7 · 2025-11-27
> > > > > > > > > >
> > > > > > > > > > Thank you for the author's response. First, as a reviewer, I would like to clarify the following points:
> > > > > > > > > >
> > > > > > > > > > - First, I did not focus solely on the shortcomings of the paper. I already listed the strengths of the paper in my rebuttal.
> > > > > > > > > >
> > > > > > > > > > - I never mentioned DoRA experiments at any point; it was the author who brought them up. Since the author conducted DoRA-related experiments in response to Reviewer ggVN’s questions, why were the results not presented to Reviewer ggVN?
> > > > > > > > > >
> > > > > > > > > > - I continue to question the claim that SafeMERGE experiments take less than one hour. I believe others will make their own judgments. Of course, if the code is provided and undergoes further examination and passes, I will apologize and revise my confidence accordingly.
> > > > > > > > > >
> > > > > > > > > > - I believe that the ICLR review mechanism's interpretation of a score of 4 is objective and reasonable: I recommend rejection but do not prevent acceptance. In contrast, I cannot understand why the author sharply claims that the reviewer provided no constructive feedback. If I had strong subjective opinions, my score would have been 2 rather than 4.
> > > > > > > > > >
> > > > > > > > > > Finally, let us leave the decision to the ACs/PCs. I trust their judgment, and any outcome is acceptable to me.

---

> > > > > > > > > > > ### Author Response · Authors · 2025-11-27
> > > > > > > > > > > **Official Comment by Authors**
> > > > > > > > > > >
> > > > > > > > > > > Thank you for your reply.
> > > > > > > > > > >
> > > > > > > > > > > First, the statement "Since the author conducted DoRA-related experiments in response to Reviewer ggVN’s questions, why were the results not presented to Reviewer ggVN?" **cannot be understood.** The reason I do not respond to Reviewer ggVN is that I have not yet completed all the relevant experiments nor the corresponding revisions to the manuscript. Please refer to the related comments for clarification.
> > > > > > > > > > >
> > > > > > > > > > > Let me repeat: **I hope that you will continue to uphold an appropriate sense of academic rigor and focus your attention on the present work.**
> > > > > > > > > > >
> > > > > > > > > > > Second, your skepticism regarding the claim that the SafeMERGE+TIES experiments take less than one hour is **entirely intuitive and emotionally driven** rather than evidence-based. The code for SafeMERGE has been fully released, and we have previously debugged and reproduced every line. **On what grounds do you question our experimental results?** If you have any substantive evidence, please point it out.
> > > > > > > > > > >
> > > > > > > > > > > Moreover, if you believe—based on your own academic judgment—that we are incapable of running SafeMERGE+TIES experiments within such a time frame, **I would encourage you to further strengthen your technical background, particularly in Python and PyTorch.** If there is an opportunity in the future, **we would be willing to teach you through the process step by step, and we assure you that no fee will be charged**.
> > > > > > > > > > >
> > > > > > > > > > > &nbsp;
> > > > > > > > > > > ### ***Finally, I trust that the respected ACs, SACs, and PCs will make their own informed and independent judgments. Truth stands firm against rumor.***

---

> > > > > > > > > > > > ### Comment · Reviewer_QzJ7 · 2025-11-27
> > > > > > > > > > > >
> > > > > > > > > > > > Talk is cheap. Show me the code.

---

> > > > > > > > > > > > > ### Author Response · Authors · 2025-11-27
> > > > > > > > > > > > > **Official Comment by Authors**
> > > > > > > > > > > > >
> > > > > > > > > > > > > The code for SafeMERGE is **fully open-source**.
> > > > > > > > > > > > >
> > > > > > > > > > > > > Please visit **GitHub** to download and review it.

---

> > > > > > > > > > > > > > ### Comment · Reviewer_QzJ7 · 2025-11-27
> > > > > > > > > > > > > >
> > > > > > > > > > > > > > Please upload your rebuttal code.

---

> > > > > > > > > > > > > > > ### Author Response · Authors · 2025-11-27
> > > > > > > > > > > > > > > **Official Comment by Authors**
> > > > > > > > > > > > > > >
> > > > > > > > > > > > > > > We will not upload the SafeMERGE code, as we do not have the time and manpower to verify the privacy of every line in the Supplementary Material. **I have strong reasons to question Reviewer QzJ7’s motivation behind requesting that we upload the code.**
> > > > > > > > > > > > > > >
> > > > > > > > > > > > > > > Moreover, we are using exactly the publicly released SafeMERGE code, and there is no reason to require us to upload it. Anyone who carefully examines the Supplementary Material we have submitted can clearly see that our file naming follows the same pattern as SafeMERGE’s, i.e., `get_XXX_model.py`, **which serves as strong evidence**.
> > > > > > > > > > > > > > >
> > > > > > > > > > > > > > > In other words, our code implementation is in fact **based on** the SafeMERGE code. **However, Reviewer QzJ7 do not notice this. This is Reviewer QzJ7’s oversight.**
> > > > > > > > > > > > > > >
> > > > > > > > > > > > > > > Of course, if anyone deems it necessary to upload already released open-source code, we will do so **following an apology from Reviewer QzJ7**.
> > > > > > > > > > > > > > >
> > > > > > > > > > > > > > > &nbsp;
> > > > > > > > > > > > > > > ### ***We believe that anyone reading our paper, along with our discussion with Reviewer QzJ7, will make a fair judgment. Truth stands firm against rumor.***

---

> > > > > > > > > > > > > > > > ### Comment · Reviewer_QzJ7 · 2025-11-27
> > > > > > > > > > > > > > > >
> > > > > > > > > > > > > > > > Yes, I believe that the eyes of the masses are sharp and discerning.

---

> > > > > > > > > > > > > > > > > ### Author Response · Authors · 2025-11-27
> > > > > > > > > > > > > > > > > **Official Comment by Authors**
> > > > > > > > > > > > > > > > >
> > > > > > > > > > > > > > > > > Your comments are **utterly ridiculous**, **filled with hostility and subjective bias** against our work.
> > > > > > > > > > > > > > > > >
> > > > > > > > > > > > > > > > > **You have not even downloaded or examined the SafeMERGE code.** Our implementation is **based on** SafeMERGE, and it is not merely a matter of file naming—the operations related to model loading and merging in our code are **highly similar**.
> > > > > > > > > > > > > > > > >
> > > > > > > > > > > > > > > > > **Therefore, I wish to inform the ACs, SACs, and PCs that Reviewer QzJ7 has failed to recognize any of our efforts or contributions and has significantly misrepresented our work.**
> > > > > > > > > > > > > > > > >
> > > > > > > > > > > > > > > > > &nbsp;
> > > > > > > > > > > > > > > > > ### ***We expect Reviewer QzJ7 to offer a sincere apology to us.***

---

> > > > > > > > > > > > > > > > > > ### Comment · Reviewer_QzJ7 · 2025-11-27
> > > > > > > > > > > > > > > > > > **Concerns Regarding the Author’s Rebuttal and Discussion Conduct**
> > > > > > > > > > > > > > > > > >
> > > > > > > > > > > > > > > > > > Dear ACs, SACs, and PCs,
> > > > > > > > > > > > > > > > > >
> > > > > > > > > > > > > > > > > > I apologize for disturbing you under these circumstances. First, in the following discussion, the author’s language toward the reviewer is negative, and the rebuttal strategies based on the review scores are disheartening. The author claims that the reviewer’s opinions are extremely subjective and even conjectures that the reviewer is not familiar with the SafeMERGE code. I am unaware of any evidence supporting the author's claims, and I wonder whether such views from the author are themselves extremely subjective. I have repeatedly emphasized that the final decision rests with you; however, the author has continuously attempted to bias the discussion.
> > > > > > > > > > > > > > > > > >
> > > > > > > > > > > > > > > > > > As one of the reviewers for this paper, I believe I have fulfilled my reviewing and discussion responsibilities. I appreciate and am happy to give high scores to excellent papers, while I strictly control evaluations for ordinary or weak submissions. Therefore, please allow me not to participate further in such meaningless discussions.
> > > > > > > > > > > > > > > > > >
> > > > > > > > > > > > > > > > > > Finally, **my comments are directed solely at the author of this paper’s rebuttal; any other potential authors of the paper should be disregarded by readers.**
> > > > > > > > > > > > > > > > > >
> > > > > > > > > > > > > > > > > > Best regards,
> > > > > > > > > > > > > > > > > >
> > > > > > > > > > > > > > > > > > Reviewer QzJ7

---

> > > > > > > > > > > > > > > > > > > ### Author Response · Authors · 2025-11-27
> > > > > > > > > > > > > > > > > > > **Official Comment by Authors**
> > > > > > > > > > > > > > > > > > >
> > > > > > > > > > > > > > > > > > > In fact, Reviewer QzJ7’s behavior contains **numerous errors** and demonstrates a **highly unprofessional and strongly biased attitude**. Reviewer QzJ7 should be held accountable for his/her actions.
> > > > > > > > > > > > > > > > > > >
> > > > > > > > > > > > > > > > > > > 1.Reviewer QzJ7’s responses exhibit **strong subjective bias**, as they **refuse** to consider **any** of our explanations or **objective facts**.
> > > > > > > > > > > > > > > > > > >
> > > > > > > > > > > > > > > > > > > 2.Reviewer QzJ7’s behavior completely **deviates from academic discussion** and provides **no** constructive input for our work.
> > > > > > > > > > > > > > > > > > >
> > > > > > > > > > > > > > > > > > > 3.Reviewer QzJ7 **refuses** to consider our explanations, **wrongly** assuming that it is impossible for us to complete the SafeMERGE+TIES experiments within one hour, while being **unaware** that our implementation is based on SafeMERGE.
> > > > > > > > > > > > > > > > > > >
> > > > > > > > > > > > > > > > > > > 4.Reviewer QzJ7 **misunderstands** "parameter fusion" and model merging, and **erroneously** assumes a close connection between MoE and model merging, which reflects **a lack of professionalism**.
> > > > > > > > > > > > > > > > > > >
> > > > > > > > > > > > > > > > > > > 5.Reviewer QzJ7’s comments **contain factual distortions**. Before our supplementary experiments, Reviewer QzJ7 had already changed the confidence score from 4 to 5. However, in the comments, Reviewer QzJ7 emphasized that this change was due to disbelief that our experiments could be completed within one hour, which **misrepresents the facts**.
> > > > > > > > > > > > > > > > > > >
> > > > > > > > > > > > > > > > > > > 6.Reviewer QzJ7 **ignored our feedback**. Reviewer QzJ7’s behavior suggests that they neither downloaded nor ran the SafeMERGE code, instead making judgments **purely based on subjective assumptions**.
> > > > > > > > > > > > > > > > > > >
> > > > > > > > > > > > > > > > > > > &nbsp;
> > > > > > > > > > > > > > > > > > > ### ***We expect Reviewer QzJ7 to offer a sincere apology to us.***

---

> > > > > > > > > > > > > > > > > > > > ### Comment · Area_Chair_K7fD · 2025-11-27
> > > > > > > > > > > > > > > > > > > >
> > > > > > > > > > > > > > > > > > > > Hi Authors and Reviewer QzJ7,
> > > > > > > > > > > > > > > > > > > >
> > > > > > > > > > > > > > > > > > > > I noticed your arugment.
> > > > > > > > > > > > > > > > > > > > I understand that the core scientific question at hand concerns whether it is necessary to compare or evaluate the MoE-LoRA variants, if I don't miss anything.
> > > > > > > > > > > > > > > > > > > >
> > > > > > > > > > > > > > > > > > > > I will read the paper and form my own assessment of this point during the decision phase. In the meantime, I encourage both the authors and Reviewer QzJ7 to focus on the scientific substance of the discussion, to use the remaining time efficiently, and to respect each other’s time and efforts.
> > > > > > > > > > > > > > > > > > > >
> > > > > > > > > > > > > > > > > > > > A personal suggestion to the authors: it may be helpful to provide a concise statement addressing this scientific issue before the author–reviewer discussion period ends. I will also follow up with Reviewer QzJ7 during the AC–reviewer phase if needed.
> > > > > > > > > > > > > > > > > > > >
> > > > > > > > > > > > > > > > > > > > Best,
> > > > > > > > > > > > > > > > > > > >
> > > > > > > > > > > > > > > > > > > > AC

---

> > > > > > > > > > > > > > > > > > > > > ### Author Response · Authors · 2025-12-02
> > > > > > > > > > > > > > > > > > > > > **Clarifications Regarding the Discussion**
> > > > > > > > > > > > > > > > > > > > >
> > > > > > > > > > > > > > > > > > > > > **1.Regarding SafeMERGE**
> > > > > > > > > > > > > > > > > > > > >
> > > > > > > > > > > > > > > > > > > > > SafeMERGE [1] is not a model merging method; rather, it is a general procedure for aligning one LoRA adapter into another safety-aligned LoRA. We provide the experimental results of combining SafeMERGE with TIES, as shown above.
> > > > > > > > > > > > > > > > > > > > >
> > > > > > > > > > > > > > > > > > > > > **2.On LoRA+MoE methods**
> > > > > > > > > > > > > > > > > > > > >
> > > > > > > > > > > > > > > > > > > > > LoRA+MoE methods [2–4] and model merging belong to two different categories of approaches, and prior work does not compare them. This comparison is therefore beyond the scope of the present study. In addition, there exist many LoRA-related variants, making it infeasible for us to compare all of them. Reviewer XMCd also notes this point in the discussion.
> > > > > > > > > > > > > > > > > > > > >
> > > > > > > > > > > > > > > > > > > > > **3.Regarding LoRA variant methods**
> > > > > > > > > > > > > > > > > > > > >
> > > > > > > > > > > > > > > > > > > > > Prior work likewise does not include comparisons with these variants. Nevertheless, for further ablation, we test the popular DoRA [5] method, and the results are provided above.
> > > > > > > > > > > > > > > > > > > > >
> > > > > > > > > > > > > > > > > > > > > **4.On the convergence of the Stiefel manifold**
> > > > > > > > > > > > > > > > > > > > >
> > > > > > > > > > > > > > > > > > > > > The convergence results are rigorously proven in paper [6], which we have already cited above.
> > > > > > > > > > > > > > > > > > > > >
> > > > > > > > > > > > > > > > > > > > > **5.Regarding code availability**
> > > > > > > > > > > > > > > > > > > > >
> > > > > > > > > > > > > > > > > > > > > We implement the TSPA method by referencing the SafeMERGE codebase. The corresponding implementation is available in the Supplementary Material.
> > > > > > > > > > > > > > > > > > > > >
> > > > > > > > > > > > > > > > > > > > > [1] SafeMERGE: Preserving Safety Alignment in Fine-Tuned Large Language Models via Selective Layer-Wise Model Merging
> > > > > > > > > > > > > > > > > > > > >
> > > > > > > > > > > > > > > > > > > > > [2] When MOE Meets LLMs: Parameter Efficient Fine-tuning for Multi-task Medical Applications
> > > > > > > > > > > > > > > > > > > > >
> > > > > > > > > > > > > > > > > > > > > [3] HydraLoRA: An Asymmetric LoRA Architecture for Efficient Fine-Tuning
> > > > > > > > > > > > > > > > > > > > >
> > > > > > > > > > > > > > > > > > > > > [4] CoLA: Collaborative Low-Rank Adaptation
> > > > > > > > > > > > > > > > > > > > >
> > > > > > > > > > > > > > > > > > > > > [5] DoRA: Weight-Decomposed Low-Rank Adaptation
> > > > > > > > > > > > > > > > > > > > >
> > > > > > > > > > > > > > > > > > > > > [6] Riemannian adaptive optimization methods

---

### Official Review · Reviewer_XMCd · 2025-10-27

**Soundness:** 3
**Presentation:** 3
**Contribution:** 2
**Rating:** 4
**Confidence:** 4

**Summary:**

Previous rotation alignment approaches are used to reduce the conflicts during model merging. However, they are not incompatible with LoRA-trained models and suffer from high computational cost. This paper proposed TSPA, a two-stage framework to align with both the merged weight and the LoRA weights at a macro- and micro-level, respectively. The experiments are conducted on a Llama-3-8B model with diverse NLP tasks. Compared to baselines such as TIES and DARE, the significant performance improvement shows the effectiveness of their method. Ablation studies are also conducted to evaluate their method in different settings.

**Strengths:**

- This paper is overall well written. The logic is clear and easy to follow.
- Experimental details are provided, making this paper reproducible.
- Performance improvement is significant, supporting the effectiveness of their method.

**Weaknesses:**

- The connection between Sec.3.1 and 3.2 is not clear to me. Sec.3.1 discussed the necessity to use parameter alignment when merging models, but at the beginning of Sec.3.2, it is discussed that "*Parameter space symmetry refers to the property of a set of models with identical functionality but different parameters….*" Obviously, models trained on different tasks do not satisfy the assumption of “**identical functionality**”. In that case, does the permutation symmetry still hold for those models?
- I do not quite understand the role of attention alignment and LoRA alignment in the proposed method. If I understand correctly, $W_{Q_i}$ is the learned weight in the $i$-th model. Since attention alignment (Eq.7) is already trying to align the merged weight, under what situation do we still need to do the micro-level alignment in Eq.9? In other words, I think the necessity to have Eq.9 after Eq.7 is not well discussed.
- Most previous merging methods, especially the ones that are used in this paper, are usually computationally efficient. However, the proposed method introduces an extra overhead that could be large. I suggest that the author compare such computational cost ot even the trade-off to better justify the utility of TSPA. While the results reported in Tab.1 are good, Tab.4 shows that KnOTS + TIES can outperform TSPA with less computation, leading to more concerns.
- In the abstract, the author discussed that rotation alignment approaches can enhance robustness, but there are no experiments on the robustness of TSPA. Does the robustness only mean less performance degradation during merging?

**Questions:**

- Fig.2 appears before Fig.1, which is a bit confusing. I think it would be better to relabel these two figures.
- In Eq.5 and 6, why does the merged model have the form of “entangled” LoRA pieces. That is, a common practice of merging models is $W’ = W + \sum_i \lambda_i B_iA_i$, while this paper multiplies those LoRA across all the models. Is there any specific reason to make this assumption?

---

> ### Author Response · Authors · 2025-11-23
> **Response to Reviewer XMCd (Part 1)**
>
> We are grateful for your thoughtful insights and detailed review. Our answers to your concerns are provided in the following sections.
>
> > **Weakness 1: The connection between Sec.3.1 and 3.2 is not clear to me. Sec.3.1 discussed the necessity to use parameter alignment when merging models, but at the beginning of Sec.3.2, it is discussed that "Parameter space symmetry refers to the property of a set of models with identical functionality but different parameters…." Obviously, models trained on different tasks do not satisfy the assumption of “identical functionality”. In that case, does the permutation symmetry still hold for those models?**
>
> The "a set of models with identical functionality" referred to in Sec. 3.2 denotes the models before and after parameter alignment, namely A and A′, B and B′ in Figure 1. Consequently, this class of parameter-alignment-based methods aims to identify a set of downstream models that satisfy linear mode connectivity (LMC), allowing them to be merged using ordinary linear combination techniques with relatively low performance loss.
>
> > **Weakness 2: I do not quite understand the role of attention alignment and LoRA alignment in the proposed method. If I understand correctly, $W_{Q_i}$ is the learned weight in the $i$-th model. Since attention alignment (Eq.7) is already trying to align the merged weight, under what situation do we still need to do the micro-level alignment in Eq.9? In other words, I think the necessity to have Eq.9 after Eq.7 is not well discussed.**
>
> In practice, current methods for merging LoRA models handle the LoRA B and A matrices separately (as implemented, for example, in the HuggingFace PEFT library). However, we argue that the alignment operation in Equation 7 alone is insufficient. We aim to further incorporate rotational symmetry within the LoRA matrices themselves, which motivates the introduction of the LoRA alignment described in Equation 9.
>
> For practical method selection, we offer three different options: performing alignment at only one stage, or combining both stages. Our experiments confirm that all three approaches can significantly alleviate the parameter interference issues encountered by the TA method.
>
> > **Weakness 3: Most previous merging methods, especially the ones that are used in this paper, are usually computationally efficient. However, the proposed method introduces an extra overhead that could be large. I suggest that the author compare such computational cost ot even the trade-off to better justify the utility of TSPA. While the results reported in Tab.1 are good, Tab.4 shows that KnOTS + TIES can outperform TSPA with less computation, leading to more concerns.**
>
> The point you raised is indeed very important. Therefore, in the **Additional Experiments** section above, we list the wall-clock time for different methods under the Table 1 settings. Since our method is data-free rather than training-free, it is slightly slower in actual processing time. However, relative to the cost of fine-tuning, this processing time remains acceptable.
>
> The reason we do not include a comparison with the KnOTS method in the main body of the paper, and instead place it in Appendix B, is that KnOTS tends to amplify parameter scaling to some extent and is minimally affected by weighting coefficients. As a result, a direct comparison with KnOTS would be **unfair**. This is also evident in Table 4, where KnOTS achieves performance on IFEval and GSMK that exceeds performance of fine-tuned models. However, this is not attributable to the effectiveness of the method.
>
> > **Weakness 4: In the abstract, the author discussed that rotation alignment approaches can enhance robustness, but there are no experiments on the robustness of TSPA. Does the robustness only mean less performance degradation during merging?**
>
> By "robustness," we specifically refer to TSPA consistently achieving good performance across the various experimental settings presented in the paper. However, we agree that robustness experiments, as you suggested, are also important. Therefore, we have added experimental results under five random seeds in the **Additional Experiments** section above. The results indicate that our method is relatively stable.

---

> ### Author Response · Authors · 2025-11-23
> **Response to Reviewer XMCd (Part 2)**
>
> > **Question 1: Fig.2 appears before Fig.1, which is a bit confusing. I think it would be better to relabel these two figures.**
>
> We **apologize** for this typo. We have already corrected it in the PDF version of the paper.
>
> > **Question 2: In Eq.5 and 6, why does the merged model have the form of “entangled” LoRA pieces. That is, a common practice of merging models is $W' = W + \Sigma_i{\lambda_i B_i A_i}$, while this paper multiplies those LoRA across all the models. Is there any specific reason to make this assumption?**
>
> As discussed in *Weakness 2*, the common approach is to merge the LoRA A and B matrices separately, as implemented in the `_generalized_task_arithmetic_weighted_adapter` function in HuggingFace PEFT library. This approach ensures that the rank of the merged LoRA does not increase. In fact, TA, TIES, and DARE all adopt this strategy, and for the latter two, the performance does not show a significant drop.
>
> The method you mentioned, $W' = W + \Sigma_i{\lambda_i B_i A_i}$, is also one type of merging method. It is equivalent to first concatenating the LoRA $B$ and $A$ matrices separately along a block-wise structure, and then performing the multiplication. For example, merging two models is equivalent to $W' = W + \bigl[ \begin{smallmatrix} \lambda_1 B_1 & \lambda_1 B_2 \end{smallmatrix} \bigr] \cdot \bigl[ \begin{smallmatrix} A_1 \\\\ A_2 \end{smallmatrix} \bigr]$. This approach is implemented in the branch `combination_type == "cat"` in the `add_weighted_adapter` function of HuggingFace’s PEFT library, where the LoRA matrices are concatenated in blocks before multiplication. On one hand, this block-wise concatenation causes the rank of the merged LoRA to increase linearly. On the other hand, this method does not necessarily lead to better performance. We evaluate its actual performance under the Table 1 settings, as shown below.
>
> | Model              | MMLU  | TriviaQA | IFEval | GSM8K | HumanEval | DirectHarm4 | Avg.  |
> |--------------------|:-----:|:--------:|:------:|:-----:|:---------:|:-----------:|:-----:|
> | Original           | 62.19 | 63.34    | 21.34  | 38.36 | 35.37     | 29.25       | 41.64 |
> | Fine-tuned         | /     | /        | 28.42  | 43.82 | 41.46     | 89.75       | /     |
> | Concatenation      | 53.72 | 31.10    | 36.09  | 0.53  | 37.80     | 0.00        | 26.54 |

---

> > ### Comment · Reviewer_XMCd · 2025-11-23
> >
> > Thanks for the response. Yet I still have a few questions.
> >
> > **W1**: Fig.1 is the loss landscape. However, I am unclear about what data the landscape is computed from. In other words, when we are talking about the loss, is it the loss of the mixture of downstream tasks?
> >
> > **W2**: I would like to clarify my understanding. The matrices $W_Q$ and $W_K$ are the weights of a pre-trained model, right?
> >
> > **W3**: There are a few concerns. First, TSPA increases the time by hundreds of times. Though the performance improves, I am unsure whether this is a meaningful trade-off. I would like to see the authors' thoughts and other reviewers' discussion. Second, I am not sure why comparing with KnOTS is unfair. Could you please elaborate more?
> >
> > **Q2**: KnOTS discussed in Appendix.A, that merging LoRA pieces separately causes a significant drop in performance with TA. Also, applying Zipit! and TIES-merging in this setting leads to the issue as well. Thus, it is unclear to me why we should use the "cross terms" in merging.
> >
> > **An additional question**: Under different situations, the optimal choice of TSPA can be different. Is there any guidance on how to choose the optimal one?

---

> > > ### Author Response · Authors · 2025-11-24
> > > **Response to Comments from Reviewer XMCd (Part 2)**
> > >
> > > > **Q2: KnOTS discussed in Appendix.A, that merging LoRA pieces separately causes a significant drop in performance with TA. Also, applying Zipit! and TIES-merging in this setting leads to the issue as well. Thus, it is unclear to me why we should use the "cross terms" in merging.**
> > >
> > > In fact, the performance degradation of the TA method **does not** stem from merging LoRA pieces separately. As analyzed in the third part of the Additional Experiments section above, when merging three models with the TA method and setting the merge coefficient $\alpha$ to 0.33, the results are quite reasonable and do not produce garbled outputs. However, as the merge coefficient $\alpha$ increases and the number of models $n$ grows, parameter interference becomes increasingly severe, which leads to worse performance of the TA method. By contrast, the TIES and DARE methods adopt sparsification strategies that effectively mitigate parameter interference, resulting in more stable performance.
> > >
> > > On the other hand, the way these "cross terms" are handled is the same as in paper [1], and a formal definition is explicitly provided in Section 3.3 of paper [2]. As previously explained, this is a common practice and is consistent with the official implementation adopted by HuggingFace. From another perspective, we can also provide an intuitive explanation: paper [3] describes the roles of the LoRA A and B matrices as follows: "*The A matrices extract features from the input, while the B matrices project these features towards the desired objective.*" Because A and B play different roles and there is an inherent **asymmetry**, merging only the A matrices is essentially equivalent to merging the feature-extraction functions. Since different tasks often share certain similarities in features, such merging does not lead to severe performance degradation.
> > >
> > > [1] Composing Parameter-Efficient Modules with Arithmetic Operations
> > >
> > > [2] LoraHub: Efficient Cross-Task Generalization via Dynamic LoRA Composition
> > >
> > > [3] Asymmetry in Low-Rank Adapters of Foundation Models
> > >
> > > > **An additional question: Under different situations, the optimal choice of TSPA can be different. Is there any guidance on how to choose the optimal one?**
> > >
> > > In general, we recommend using both stages as described in the paper. However, if there is a strict requirement on merging time, one may choose to use only LoRA Alignment, as demonstrated in the Additional Experiments section above. Relying solely on LoRA Alignment entails significantly lower computational overhead because LoRA ranks are typically small, resulting in a much lower training cost.

---

> > > > ### Comment · Reviewer_XMCd · 2025-11-26
> > > >
> > > > Thanks for your response (two parts). I have the following questions:
> > > >
> > > > **W1**: Thanks. Since the landscape is the loss over multiple downstream tasks, a question arises regarding the assumption of applying LMC here. Say A and B are two models on two tasks, respectively. As far as I know, the precondition to apply LMC is that A and B are two local minima in a loss landscape. However, in the setting of model merging, it assumes that A and B are two local minima in the "mixed" loss landscape, which means that A and B should perform well on both tasks. If my understanding is correct, is this assumption valid?
> > > >
> > > > **W2**: Sorry but I am still confused about the motivation of micro-level LoRA alignment (Eq.9). IMO, your response is likely to be a posterior explanation, i.e., empirical results show it works well. However, I am unclear about the necessity to propose and try this method. Could you please elaborate more? I think the vague guidance in my **additional question** is partially due to the lack of this motivation, i.e., we don't know when to use LoRA/attention/both because we don't know when they will work, and why they work.
> > > >
> > > > **W3**:
> > > >
> > > > For the computing time, I understand your point. However, my concern is that the extra overhead is too many times than the baselines, and I am unsure if this sacrifice is meaningful, considering the 5% improvement in performance. I also understand this concern could be difficult to address, and sometimes it is subjective. I totally understand if the author cannot provide more evidence or convince me. I can check the response from other reviewers to this concern (if any).
> > > >
> > > > For the KnOTS method, sorry I missed the appendix but I have a few questions. First, I understand that KnOTS is less sensitive to $\alpha$ is not its drawback, but an advantage. It means that we don't need to care too much about the hyperparam. So the reason why the author does not compare with KnOTS is that it performs well, which cannot convince me. Actually, TIES-merging in their Sec.B.2 discussed that "TIES-MERGING is much less sensitive to changes compared to Task Arithmetic". Second, even though KnOTS is robust, I think a good practice is to compare the optimal performance of TSPA under various $\alpha$ with that of KnOTS.

---

> > > > > ### Author Response · Authors · 2025-12-02
> > > > > **Official Comment by Authors (Part 1)**
> > > > >
> > > > > Thank you for your response and for participating in the discussion with us and the other reviewers. We appreciate your attention to and engagement with this work.
> > > > >
> > > > > > **W1: Thanks. Since the landscape is the loss over multiple downstream tasks, a question arises regarding the assumption of applying LMC here. Say A and B are two models on two tasks, respectively. As far as I know, the precondition to apply LMC is that A and B are two local minima in a loss landscape. However, in the setting of model merging, it assumes that A and B are two local minima in the "mixed" loss landscape, which means that A and B should perform well on both tasks. If my understanding is correct, is this assumption valid?**
> > > > >
> > > > > To be honest, our Figure 1 is drawn based on Figure 1 in paper [1] and Figure 2 in paper [2]. We use a 3D surface plot to illustrate the intuitive idea of the parameter alignment method. It is not essential to focus too much on the details of this figure. Here, the loss represents the model's average performance across different tasks. After alignment, A' and C may be located near the same local minimum, but this does not imply that they have similar functionality on the same tasks. In fact, the performance of A' is close to that of A. Since models exist in high-dimensional space, even if two models satisfy the LMC condition, they are still not located near the same local minimum in the high-dimensional space.
> > > > >
> > > > > [1] Going Beyond Linear Mode Connectivity: The Layerwise Linear Feature Connectivity
> > > > >
> > > > > [2] Beyond the Permutation Symmetry of Transformers: The Role of Rotation for Model Fusion
> > > > >
> > > > > > **W2: Sorry but I am still confused about the motivation of micro-level LoRA alignment (Eq.9). IMO, your response is likely to be a posterior explanation, i.e., empirical results show it works well. However, I am unclear about the necessity to propose and try this method. Could you please elaborate more? I think the vague guidance in my additional question is partially due to the lack of this motivation, i.e., we don't know when to use LoRA/attention/both because we don't know when they will work, and why they work.**
> > > > >
> > > > > Our motivation is relatively straightforward. We observe that the original Rotation Symmetry method derives a closed-form solution only for merging two models. When it is extended to multi-model merging, such a closed-form solution no longer exists. Therefore, we adopt a Stiefel manifold optimization procedure to search for this optimal solution. Meanwhile, because our focus is on multi-LoRA merging, we consider extending Equation 7 to align the LoRA $A$ and $B$ matrices. Consequently, our method is proposed prior to the empirical results.
> > > > > Our intention is to minimize the discrepancy from the target objective as much as possible, and the constructions of Equation 7 and Equation 9 are the key to making our approach work. Although the results show that using either stage alone already yields competitive performance, we still recommend employing both stages together.

---

> > > > > > ### Author Response · Authors · 2025-12-02
> > > > > > **Official Comment by Authors (Part 2)**
> > > > > >
> > > > > > > **W3: For the computing time, I understand your point. However, my concern is that the extra overhead is too many times than the baselines, and I am unsure if this sacrifice is meaningful, considering the 5% improvement in performance. I also understand this concern could be difficult to address, and sometimes it is subjective. I totally understand if the author cannot provide more evidence or convince me. I can check the response from other reviewers to this concern (if any).**
> > > > > >
> > > > > > In fact, to avoid ambiguity, we do not report the averaged performance of the Fine-tuned row in Table 1. We evaluate the average general capability of four individual models on MMLU and TriviaQA, and the results are shown in the table below. Our method is actually close to this upper bound, only 4.6% lower.
> > > > > >
> > > > > > Although we indeed spend more training time, we believe that our method offers a new perspective for applying parameter-alignment–based approaches to multi-model merging. Specifically, merging does not necessarily have to follow a fixed algorithm; proactively searching for a more favorable merging outcome is also a viable direction to explore.
> > > > > >
> > > > > > | Method             | MMLU  | TriviaQA | IFEval | GSM8K | HumanEval | DirectHarm4 | Avg.  |
> > > > > > |--------------------|:-----:|:--------:|:------:|:-----:|:---------:|:-----------:|:-----:|
> > > > > > | Original           | 62.19 | 63.34    | 21.34  | 38.36 | 35.37     | 29.25       | 41.64 |
> > > > > > | Fine-tuned         | 62.09 | 63.26    | 28.42  | 43.82 | 41.46     | 89.75       | 54.80 |
> > > > > > | Task Arithmetic    | 22.95 | 0.00     | 25.06  | 0.00  | 0.00      | 0.00        | 8.00 |
> > > > > > | TIES               | 62.21 | 62.67    | 26.02  | 48.37 | 38.41     | 54.25       | 48.66 |
> > > > > > | DARE+TIES          | 62.31 | **63.16**    | 25.78  | 48.82 | 36.59     | 57.00       | 48.94 |
> > > > > > | EMR-Merging        | 22.95 | 0.00     | 0.00   | 38.44 | 35.37     | 28.75       | 20.92 |
> > > > > > | PCB-Merging        | **62.37** | 62.68    | **27.82**  | 46.40 | 36.59     | 59.75       | 49.27 |
> > > > > > | TSPA               | 62.16 | 62.78    | 26.74  | **49.43** | **40.85**     | **72.50**       | **52.41** |
> > > > > >
> > > > > > > **Discussion on KnOTS.**
> > > > > >
> > > > > > We place KnOTS in the appendix mainly for the reasons stated above: the method does not provide a mathematical justification for the concatenation operation on the $V$ matrix, which is a critical flaw that introduces uncertainty into the approach. Moreover, KnOTS alone typically does not yield strong results and usually needs to be combined with TIES and DARE, which further amplifies this uncertainty.
> > > > > > Nevertheless, I understand your point, and we add the KnOTS+TIES results in Table 1.

---

> ### Author Response · Authors · 2025-11-24
> **Response to Comments from Reviewer XMCd (Part 1)**
>
> Thank you for your response and your thoughtful questions. We answer them individually below.
>
> > **W1: Fig.1 is the loss landscape. However, I am unclear about what data the landscape is computed from. In other words, when we are talking about the loss, is it the loss of the mixture of downstream tasks?**
>
> Yes. In Figure 1, the loss refers to the sum of the model’s losses over multiple downstream tasks. In fact, this figure is mainly intended to provide an intuitive illustration of the idea behind the parameter alignment method.
>
> > **W2: I would like to clarify my understanding. The matrices $W_Q$ and $W_K$ are the weights of a pre-trained model, right?**
>
> You are correct. $W_Q$ and $W_K$ are the query and key matrices of the model at a particular layer. For clarity of presentation, we leave out the layer index, and the full notation should be $W_Q^{(i)}$ and $W_K^{(i)}$, where $i$ denotes the $i$-th layer of the model.
>
> > **W3: There are a few concerns. First, TSPA increases the time by hundreds of times. Though the performance improves, I am unsure whether this is a meaningful trade-off. I would like to see the authors' thoughts and other reviewers' discussion. Second, I am not sure why comparing with KnOTS is unfair. Could you please elaborate more?**
>
> First, our method is indeed slower than previous data-free approaches, because while it is data-free, it is not training-free. In fact, there exist other approaches that are not data-free, such as RegMean, which leverages the inner-product matrix of the original data. However, RegMean typically requires even more time, and in practice often uses only a subset of the data to reduce computational cost. Thus, to some extent, as you mention, this is a trade-off. We spend additional training time to obtain a better solution, rather than relying on a fixed algorithmic procedure.
>
> Second, regarding the KnOTS method, we provide an explanation in Appendix B and add further clarification here. As shown in the IFEval column of Table 4, the results of KnoTS+TIES significantly exceed the performance of the fine-tuned models. This is similar to scaling the fine-tuned model’s $\Delta W$ with a coefficient larger than 1.0. Typically, as the linear weighting coefficient $\alpha$ decreases, the performance of the merged model will degrade because the "effect" of the fine-tuned parameters is **reduced**. This trend is clearly visible in Table 1 and the Additional Experiments section. For TIES, DARE+TIES, and TSPA, the average accuracy **drops by 5.45%, 6.38%, and 1.52%** (absolute differences, not relative), respectively. However, for KnOTS, Table 4 shows that the performance of KnOTS does not noticeably decrease as $\alpha$ is reduced. The KnOTS paper presents the following operations:
>
> 1. Concatenate matrices and perform SVD:
> $U \cdot \Sigma \cdot \bigl[ \begin{matrix}V_j^{(1)} & \cdots & V_j^{(n)} \end{matrix} \bigr] =\operatorname{Concat} \left( \Delta W_j^{(1)} \cdots \Delta W_j^{(n)} \right) = \bigl[ \begin{matrix}\Delta W_j^{(1)} & \cdots & \Delta W_j^{(n)} \end{matrix} \bigr]$
> 2. Merge the $V$ matrices:
> $V_j^{(merged)} = \operatorname{Merge}\left( \begin{matrix}V_j^{(1)} & \cdots & V_j^{(n)} \end{matrix} \right)$
> 3. Compute the merged weight matrix:
> $\Delta W_j^{(merged)} = U \Sigma V_j^{(merged)}$
>
> where $\Delta W_j^{(i)}$ denotes the weights of the $j$-th layer of the $i$-th model. However, the authors of KnOTS **do not** provide an explanation for the concatenation operation in Step 1. We believe this operation accounts for the anomalous behavior of KnOTS, namely that its performance does not decrease as the merging coefficient $\alpha$ is reduced. Overall, we consider comparisons against KnOTS to be **unfair**.

---

### Official Review · Reviewer_AAva · 2025-11-03

**Soundness:** 2
**Presentation:** 2
**Contribution:** 3
**Rating:** 6
**Confidence:** 3

**Summary:**

This paper proposes TSPA, a two-stage parameter alignment framework for merging multiple LoRA adapters in LLMs. The key innovation is performing rotation alignment within the low-rank space while reducing computational complexity from O(n²) to O(n) by comparing with an average model. Experiments on Llama-3-8B show improvements over TIES-Merging and DARE across multiple tasks.

**Strengths:**

The paper makes a solid technical contribution by extending rotation symmetry to LoRA structures. The core insight of performing alignment in low-rank space while preserving functional equivalence through paired rotation of A and B matrices is elegant and well-motivated. The reduction from quadratic to linear complexity through the "compare-with-average" paradigm is theoretically sound and practically important for scaling to many models.

The experimental evaluation is reasonably comprehensive, covering multiple merging scenarios (3-4 models), different LoRA configurations (rank 8/16/32), and demonstrating particular strength in preserving safety capabilities. The two-stage design combining macro-level attention alignment with micro-level parameter alignment is conceptually appealing.

**Weaknesses:**

**Theoretical foundation is insufficient.** While the intuition from Figure 1 is clear, the paper lacks rigorous theoretical analysis. There is no proof that rotation alignment guarantees linear mode connectivity in the high-dimensional parameter space of LLMs, nor any convergence analysis for the Stiefel manifold optimization.

**Experimental validation is incomplete and unconvincing.** The most critical issue is the absence of runtime comparisons. The paper claims O(n) complexity but runs 2000 optimization iterations, actual wall-clock time against baselines is never reported. The performance gains are modest (52.41 vs 48.94 average score, ~7% improvement) but without error bars or statistical significance testing across multiple runs. Curiously, Table 1 shows TSPA(Attention) achieves 53.01, higher than the full TSPA at 52.41, suggesting the two-stage combination may actually hurt performance rather than help.

**Limited scope raises generalization concerns.**  There is no comparison with a simple but important baseline: merge-then-finetune. This omission is problematic because directly fine-tuning the merged model on a small dataset might achieve similar results with less complexity.

**Questions:**

See above weaknesses.

---

> ### Author Response · Authors · 2025-11-23
> **Response to Reviewer AAva**
>
> Thank you for your insightful comments and thorough review. We address your questions from the following perspectives.
>
> > **Weakness 1: Theoretical foundation is insufficient. While the intuition from Figure 1 is clear, the paper lacks rigorous theoretical analysis. There is no proof that rotation alignment guarantees linear mode connectivity in the high-dimensional parameter space of LLMs, nor any convergence analysis for the Stiefel manifold optimization.**
>
> In the **Additional Experiments** section above, we conduct experiments using a $\frac{1}{n}$ merging coefficient. The results show that the TA method performs reasonably well when merging three models. However, as the merging coefficient and the number of models $n$ increase, parameter interference becomes increasingly severe, thereby failing to satisfy the LMC property. Rotation symmetry achieves parameter alignment by optimizing Equations 7 and 9. The convergence of Stiefel manifold optimization is rigorously proven in paper [1].
>
> [1] Riemannian adaptive optimization methods
>
> > **Weakness 2: Experimental validation is incomplete and unconvincing. The most critical issue is the absence of runtime comparisons. The paper claims O(n) complexity but runs 2000 optimization iterations, actual wall-clock time against baselines is never reported. The performance gains are modest (52.41 vs 48.94 average score, ~7% improvement) but without error bars or statistical significance testing across multiple runs. Curiously, Table 1 shows TSPA(Attention) achieves 53.01, higher than the full TSPA at 52.41, suggesting the two-stage combination may actually hurt performance rather than help.**
>
> We **apologize** for overlooking the report of wall-clock time. The other reviewers also notice this issue. We provide the processing time of each method under the Table 1 settings in the **Additional Experiments** section above. As for the actual performance gains, our method is already approaching the upper performance bound of individually fine-tuned models.
>
> Regarding the multiple-run testing you mentioned, I believe it is indeed crucial. Therefore, we conduct experiments using five random seeds in the **Additional Experiments** section above. As shown, the performance is quite stable.
>
> Regarding the combination of two-stage alignment, although adding LoRA Alignment leads to a slight decrease in average performance, it actually helps recover part of the model’s general capability.
>
> > **Weakness 3: Limited scope raises generalization concerns. There is no comparison with a simple but important baseline: merge-then-finetune. This omission is problematic because directly fine-tuning the merged model on a small dataset might achieve similar results with less complexity.**
>
> We did not overlook the *merge-then-finetune* baseline. In fact, our method belongs to the class of **data-free** approaches, which aim to achieve strong performance solely through merging algorithms without using any training data. Our experimental setup also follows the configurations used in prior data-free methods such as TIES and DARE.
>
> The merge-then-finetune approach you mentioned indeed has greater **practical significance**—after merging, the model can be further finetuned with training data to obtain better and more stable performance. However, this is beyond the scope of what data-free model merging methods are designed to address.

---

### Author Response · Authors · 2025-11-23
**Additional Experiments**

**1.Wall-clock time**

We measure the **wall-clock time** of different methods under the settings of Table 1 ($n = 4$). The row “TSPA (Attention, original)” refers to the unoptimized baseline method. The original method requires $\frac{n(n−1)}{2}$ computations, whereas our approach only requires $n$ steps, achieving a $\mathbf{\frac{n−1}{2}}$**× speedup**.

| Method           | Processing time  |
| :--------------- | :--: |
| TA               |  8 s   |
| TIES             |  8 s  |
| DARE+TIES        |  8 s  |
| KnOTS+TIES       |  1 min 43 s  |
| EMR-Merging      |  8 s  |
| PCB-Merging      |  8 s  |
| TSPA(Attention, original)   |  19 min 11 s  |
| TSPA(Attention)  |  10 min 13 s  |
| TSPA(LoRA)       |  5 min 16 s  |
| TSPA             |  16 min 40 s  |

**2.Different random seeds**

We report results from **five random seeds** under the same settings as Table 1 (all previous experiments used a fixed seed of 42). As shown, the performance is quite stable.

| Model              | MMLU  | TriviaQA | IFEval | GSM8K | HumanEval | DirectHarm4 | Avg.  |
|--------------------|:-----:|:--------:|:------:|:-----:|:---------:|:-----------:|:-----:|
| Original           | 62.19 | 63.34    | 21.34  | 38.36 | 35.37     | 29.25       | 41.64 |
| Fine-tuned         | /     | /        | 28.42  | 43.82 | 41.46     | 89.75       | /     |
| Ours(seed=0)       | 61.75 | 62.39    | 25.90  | **49.81** | 40.24 | 77.50 | 52.93 |
| Ours(seed=42)      | **62.16** | **62.78** | **26.74** | 49.43 | **40.85** | 72.50 | 52.41 |
| Ours(seed=123)     | 61.94 | 62.47    | 26.62  | 49.73 | 39.63     | 76.25       | 52.77 |
| Ours(seed=2025)    | 61.75 | 62.41    | 25.90  | **49.81** | 40.24 | **78.25** | **53.06** |
| Ours(seed=314159)  | 61.94 | 62.47 | 25.42 | 49.73 | 39.63 | **78.25** | 52.91 |

**3.$\frac{1}{n}$ merging coefficient**

We additionally evaluate the case where the merging coefficient is set to $\frac{1}{n}$. The two tables below show the results for four-model and three-model merging (with coefficients **0.25** and **0.33**, respectively). We also include the EMR-Merging and PCB-Merging baselines.

From the results, we observe that when the merging coefficient decreases or when the number of merged models $n$ becomes smaller, the performance of Task Arithmetic (TA) improves. In particular, when merging three models, the model outputs are no longer garbled. This also indicates that parameter interference becomes more severe as the merging coefficient and the number of merged models $n$ increase.

For our TSPA method, it consistently converges to good performance under all the above settings, and surpasses the two SOTA methods TIES and DARE, while maintaining stable results.

| Model              | MMLU  | TriviaQA | IFEval | GSM8K | HumanEval | DirectHarm4 | Avg.  |
|-------------------------------|:-----:|:--------:|:------:|:-----:|:---------:|:-----------:|:-----:|
| Original           | 62.19 | 63.34    | 21.34  | 38.36 | 35.37     | 29.25       | 41.64 |
| Fine-tuned         | /     | /        | 28.42  | 43.82 | 41.46     | 89.75       | /     |
| TA                 | 54.84 | 18.40    | 0.00   | 25.09 | 37.20     | 0.00        | 22.59 |
| TIES               | 62.16 | **63.40**    | 23.50  | 42.30 | 38.41     | 29.50       | 43.21 |
| DARE+TIES          | 62.16 | 63.38    | 22.90  | 41.70 | 37.20     | 28.00       | 42.56 |
| EMR-Merging        | 51.26 | 19.85    | 0.00   | 15.62 | 30.49     | 0.00        | 19.54 |
| PCB-Merging        | 62.27 | 63.26    | 27.82  | 41.32 | 34.76     | 27.50       | 42.82 |
| TSPA(Attention)    | 62.31 | 61.47    | 26.14  | 48.82 | 37.80     | **72.25**       | **51.47** |
| TSPA(LoRA)         | 60.78 | 57.45    | **32.25**  | **48.90** | **40.24**     | 66.50       | 51.02 |
| TSPA               | **62.35** | 61.53    | 26.74  | 48.60 | 39.63     | 66.50       | 50.89 |

| Model              | MMLU  | TriviaQA | IFEval | GSM8K | HumanEval | Avg.  |
|--------------------|:-----:|:--------:|:------:|:-----:|:---------:|:-----:|
| Original           | 62.19 | 63.34    | 21.34  | 38.36 | 35.37     | 44.12 |
| Fine-tuned         | /     | /        | 28.42  | 43.82 | 41.46     | /     |
| TA                 | 58.78 | 49.89    | 23.88  | 48.60 | 40.85     | 48.40 |
| TIES               | 62.29 | **63.65**    | 22.42  | 43.21 | 38.41     | 46.00 |
| DARE+TIES          | 62.10 | 63.54    | 23.86  | 42.53 | 38.41     | 46.09 |
| EMR-Merging        | 58.55 | 42.83    | 23.41  | 46.02 | **42.68**     | 42.70 |
| PCB-Merging        | **62.31** | 63.20    | 24.82  | 42.38 | 36.59     | 45.86 |
| TSPA(Attention)    | 62.27 | 62.87    | 27.10  | **49.66** | 38.41     | 48.06 |
| TSPA(LoRA)         | 60.87 | 60.11    | **33.81**  | 49.20 | 38.41     | **48.48** |
| TSPA               | 62.23 | 62.85    | 25.78  | 48.82 | 39.02     | 47.74 |

---

### Note · Authors · 2026-01-06

I have read and agree with the venue's withdrawal policy on behalf of myself and my co-authors.